# CONTINUOUS-TIME META-LEARNING WITH FORWARD MODE DIFFERENTIATION

**Tristan Deleu**[*]    **David Kanaa**    **Leo Feng**    **Giancarlo Kerg**
**Yoshua Bengio**[1,2]    **Guillaume Lajoie**[2]    **Pierre-Luc Bacon**[2]

Mila – Université de Montréal

## ABSTRACT

Drawing inspiration from gradient-based meta-learning methods with infinitely small gradient steps, we introduce Continuous-Time Meta-Learning (COMLN), a meta-learning algorithm where adaptation follows the dynamics of a gradient vector field. Specifically, representations of the inputs are meta-learned such that a task-specific linear classifier is obtained as a solution of an ordinary differential equation (ODE). Treating the learning process as an ODE offers the notable advantage that the length of the trajectory is now continuous, as opposed to a fixed and discrete number of gradient steps. As a consequence, we can optimize the amount of adaptation necessary to solve a new task using stochastic gradient descent, in addition to learning the initial conditions as is standard practice in gradient-based meta-learning. Importantly, in order to compute the exact meta-gradients required for the outer-loop updates, we devise an efficient algorithm based on forward mode differentiation, whose memory requirements do not scale with the length of the learning trajectory, thus allowing longer adaptation in constant memory. We provide analytical guarantees for the stability of COMLN, we show empirically its efficiency in terms of runtime and memory usage, and we illustrate its effectiveness on a range of few-shot image classification problems.

## 1 INTRODUCTION

Among the existing meta-learning algorithms, gradient-based methods as popularized by Model-Agnostic Meta-Learning (MAML, Finn et al., 2017) have received a lot of attention over the past few years. They formulate the problem of learning a new task as an inner optimization problem, typically based on a few steps of gradient descent. An outer *meta*-optimization problem is then responsible for updating the meta-parameters of this learning process, such as the initialization of the gradient descent procedure. However since the updates at the outer level typically require backpropagating through the learning process, this class of methods has often been limited to only a few gradient steps of adaptation, due to memory constraints. Although solutions have been proposed to alleviate the memory requirements of these algorithms, including checkpointing (Baranchuk, 2019), using implicit differentiation (Rajeswaran et al., 2019), or reformulating the meta-learning objective (Flennerhag et al., 2018), they are generally either more computationally demanding, or only approximate the gradients of the meta-learning objective (Nichol et al., 2018; Flennerhag et al., 2020).

In this work, we propose a continuous-time formulation of gradient-based meta-learning, called *Continuous-Time Meta-Learning* (COMLN), where the adaptation is the solution of a differential equation (see Figure 1). Moving to continuous time allows us to devise a novel algorithm, based on forward mode differentiation, to efficiently compute the exact gradients for meta-optimization, no matter how long the adaptation to a new task might be. We show that using forward mode differentiation leads to a stable algorithm, unlike the counterpart of backpropagation in continuous time called the adjoint method (frequently used in the Neural ODE literature; Chen et al., 2018) which tends to be unstable in conjunction with gradient vector fields. Moreover as the length of

---

[*]Correspondence to: Tristan Deleu <deleutri@mila.quebec>
[1]CIFAR Senior Fellow, [2]CIFAR AI Chair
  Code is available at: https://github.com/tristandeleu/jax-comln

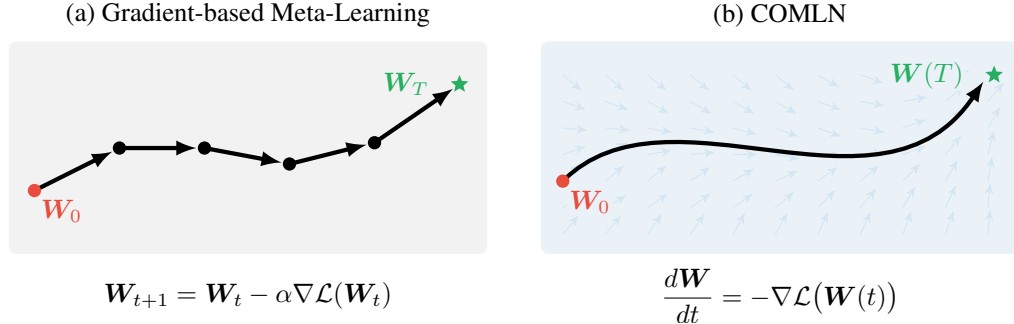

(a) Gradient-based Meta-Learning

$$W_{t+1} = W_t - \alpha \nabla \mathcal{L}(W_t)$$

(b) COMLN

$$\frac{dW}{dt} = -\nabla \mathcal{L}(W(t))$$

Figure 1: Illustration of the adaptation process in (a) a gradient-based meta-learning algorithm, such as ANIL (Raghu et al., 2019), where the adapted parameters $W_T$ are given after $T$ steps of gradient descent, and in (b) Continuous-Time Meta-Learning (COMLN), where the adapted parameters $W(T)$ are the result of following the dynamics of a gradient vector field up to time $T$.

the adaptation trajectory is a continuous quantity, as opposed to a discrete number of gradient steps fixed ahead of time, we can treat the amount of adaptation in COMLN as a meta-parameter—on par with the initialization—which we can meta-optimize using stochastic gradient descent. We verify empirically that our method is both computationally and memory efficient, and we show that COMLN outperforms other standard meta-learning algorithms on few-shot image classification datasets.

## 2 BACKGROUND

In this work, we consider the problem of few-shot classification, that is the problem of learning a classification model with only a small number of training examples. More precisely for a classification task $\tau$, we assume that we have access to a (small) training dataset $\mathcal{D}_\tau^{\text{train}} = \{(\boldsymbol{x}_m, \boldsymbol{y}_m)\}_{m=1}^M$ to fit a model on task $\tau$, and a distinct test dataset $\mathcal{D}_\tau^{\text{test}}$ to evaluate how well this adapted model generalizes on that task. In the few-shot learning literature, it is standard to consider the problem of $k$-shot $N$-way classification, meaning that the model has to classify among $N$ possible classes, and there are only $k$ examples of each class in $\mathcal{D}_\tau^{\text{train}}$, so that overall the number of training examples is $M = kN$. We use the convention that the target labels $\boldsymbol{y}_m \in \{0, 1\}^N$ are one-hot vectors.

### 2.1 GRADIENT-BASED META-LEARNING

Gradient-based meta-learning methods aim to learn an initialization such that the model is able to adapt to a new task via gradient descent. Such methods are often cast as a bi-level optimization process: adapting the task-specific parameters $\boldsymbol{\theta}$ in the inner loop, and training the (task-agnostic) meta-parameters $\Phi$ and initialization $\boldsymbol{\theta}_0$ in the outer loop. The meta-learning objective is:

$$\min_{\boldsymbol{\theta}_0, \Phi} \ \mathbb{E}_\tau \left[ \mathcal{L}(\boldsymbol{\theta}_T^\tau, \Phi; \mathcal{D}_\tau^{\text{test}}) \right] \tag{1}$$

$$\text{s.t. } \boldsymbol{\theta}_{t+1}^\tau = \boldsymbol{\theta}_t^\tau - \alpha \nabla_{\boldsymbol{\theta}} \mathcal{L}(\boldsymbol{\theta}_t^\tau, \Phi; \mathcal{D}_\tau^{\text{train}}) \qquad \boldsymbol{\theta}_0^\tau = \boldsymbol{\theta}_0 \qquad \forall \tau \sim p(\tau), \tag{2}$$

where $T$ is the number of inner loop updates. For example, in the case of MAML (Finn et al., 2017), there is no additional meta-parameter other than the initialization ($\Phi \equiv \emptyset$); in ANIL (Raghu et al., 2019), $\boldsymbol{\theta}$ are the parameters of the last layer, and $\Phi$ are the parameters of the shared embedding network; in CAVIA (Zintgraf et al., 2019), $\boldsymbol{\theta}$ are referred to as context parameters.

During meta-training, the model is trained over many tasks $\tau$. The task-specific parameters $\boldsymbol{\theta}$ are learned via gradient descent on $\mathcal{D}_\tau^{\text{train}}$. The meta-parameters are then updated by evaluating the error of the trained model on the test dataset $\mathcal{D}_\tau^{\text{test}}$. At meta-testing time, the meta-trained model is adapted on $\mathcal{D}_\tau^{\text{train}}$—i.e. applying (2) with the learned meta-parameters $\boldsymbol{\theta}_0$ and $\Phi$.

## 2.2 LOCAL SENSITIVITY ANALYSIS OF ORDINARY DIFFERENTIAL EQUATIONS

Consider the following (autonomous) Ordinary Differential Equation (ODE):

$$\frac{d\boldsymbol{z}}{dt} = g\big(\boldsymbol{z}(t); \boldsymbol{\theta}\big) \qquad\qquad \boldsymbol{z}(0) = \boldsymbol{z}_0(\boldsymbol{\theta}), \qquad\qquad (3)$$

where the dynamics $g$ and initial value $\boldsymbol{z}_0$ may depend on some external parameters $\boldsymbol{\theta}$, and integration is carried out from $0$ to some time $T$. *Local sensitivity analysis* is the study of how the solution of this dynamical system responds to local changes in $\boldsymbol{\theta}$; this effectively corresponds to calculating the derivative $d\boldsymbol{z}(t)/d\boldsymbol{\theta}$. We present here two methods to compute this derivative, with a special focus on their memory efficiency.

**Adjoint sensitivity method**  Based on the adjoint state (Pontryagin, 2018), and taking its root in control theory (Lions & Magenes, 2012), the *adjoint sensitivity method* (Bryson & Ho, 1969; Chavent et al., 1974) provides an efficient approach for evaluating derivatives of $\mathcal{L}\big(\boldsymbol{z}(T); \boldsymbol{\theta}\big)$, a function of $\boldsymbol{z}(T)$ the solution of the ODE in (3). This method, popularized lately by the literature on Neural ODEs (Chen et al., 2018), requires the integration of the adjoint equation

$$\frac{d\boldsymbol{a}}{dt} = -\boldsymbol{a}(t)\frac{\partial g\big(\boldsymbol{z}(t); \boldsymbol{\theta}\big)}{\partial \boldsymbol{z}(t)} \qquad\qquad \boldsymbol{a}(T) = \frac{d\mathcal{L}\big(\boldsymbol{z}(T); \boldsymbol{\theta}\big)}{d\boldsymbol{z}(T)}, \qquad\qquad (4)$$

backward in time. The adjoint sensitivity method can be viewed as a continuous-time counterpart to backpropagation, where the forward pass would correspond to integrating (3) forward in time from $0$ to $T$, and the backward pass to integrating (4) backward in time from $T$ to $0$.

One possible implementation, reminiscent of backpropagation through time (BPTT), is to store the intermediate values of $\boldsymbol{z}(t)$ during the forward pass, and reuse them to compute the adjoint state during the backward pass. While several sophisticated checkpointing schemes have been proposed (Serban & Hindmarsh, 2003; Gholami et al., 2019), with different compute/memory trade-offs, the memory requirements of this approach typically grow with $T$; this is similar to the memory limitations standard gradient-based meta-learning methods suffer from as the number of gradient steps increases. An alternative is to augment the adjoint state $\boldsymbol{a}(t)$ with the original state $\boldsymbol{z}(t)$, and to solve this augmented dynamical system backward in time (Chen et al., 2018). This has the notable advantage that the memory requirements are now independent of $T$, since $\boldsymbol{z}(t)$ are no longer stored during the forward pass, but they are recomputed on the fly during the backward pass.

**Forward sensitivity method**  While the adjoint method is related to reverse-mode automatic differentiation (backpropagation), the *forward sensitivity method* (Feehery et al., 1997; Leis & Kramer, 1988; Maly & Petzold, 1996; Caracotsios & Stewart, 1985), on the other hand, can be viewed as the continuous-time counterpart to forward (tangent-linear) mode differentiation (Griewank & Walther, 2008). This method is based on the fact that the derivative $\mathcal{S}(t) \triangleq d\boldsymbol{z}(t)/d\boldsymbol{\theta}$ is the solution of the so-called *forward sensitivity equation*

$$\frac{d\mathcal{S}}{dt} = \frac{\partial g\big(\boldsymbol{z}(t); \boldsymbol{\theta}\big)}{\partial \boldsymbol{z}(t)}\mathcal{S}(t) + \frac{\partial g\big(\boldsymbol{z}(t); \boldsymbol{\theta}\big)}{\partial \boldsymbol{\theta}} \qquad\qquad \mathcal{S}(0) = \frac{\partial \boldsymbol{z}_0}{\partial \boldsymbol{\theta}}. \qquad\qquad (5)$$

This equation can be found throughout the literature in optimal control and system identification (Betts, 2010; Biegler, 2010). Unlike the adjoint method, which requires an explicit forward *and* backward pass, the forward sensitivity method only requires the integration forward in time of the original ODE in (3), augmented by the sensitivity state $\mathcal{S}(t)$ with the dynamics above. The memory requirements of the forward sensitivity method do not scale with $T$ either, but it now requires storing $\mathcal{S}(t)$, which may be very large; we will come back to this problem in Section 3.2. We will simply note here that in discrete-time, this is the same issue afflicting forward-mode training of RNNs with real-time recurrent learning (RTRL; Williams & Zipser, 1989), or other meta-learning algorithms (Sutton, 1992; Franceschi et al., 2017; Xu et al., 2018).

## 3  CONTINUOUS-TIME ADAPTATION

In the limit of infinitely small steps, some optimization algorithms can be viewed as the solution trajectory of a differential equation. This point of view has often been taken to analyze their behavior

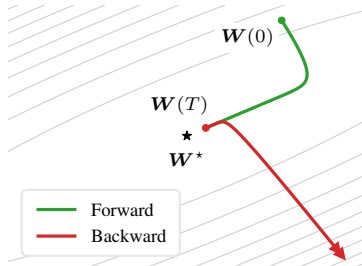

Figure 2: Numerical instability of the adjoint method applied to the gradient vector field of a quadratic loss function. The trajectory in green starting at $\boldsymbol{W}(0)$ corresponds to the integration of the dynamical system in (8) forward in time up to $T$, and the trajectory in red starting at $\boldsymbol{W}(T)$ corresponds to its integration backward in time. Note that $T$ was chosen so that $\boldsymbol{W}(T)$ does not reach the equilibrium/minimum of the loss $\boldsymbol{W}^{\star}$.

(Platt & Barr, 1988; Wilson et al., 2016; Su et al., 2014; Orvieto & Lucchi, 2019). In fact, some optimization algorithms such as gradient descent with momentum have even been introduced initially from the perspective of dynamical systems (Polyak, 1964). As the simplest example, gradient descent with a constant step size $\alpha \to 0^{+}$ (i.e. $\alpha$ tends to 0 by positive values) corresponds to following the dynamics of an autonomous ODE called a *gradient vector field*

$$\boldsymbol{z}_{t+1} = \boldsymbol{z}_t - \alpha \nabla f(\boldsymbol{z}_t) \qquad \xrightarrow[\alpha \to 0^{+}]{} \qquad \frac{d\boldsymbol{z}}{dt} = -\nabla f\big(\boldsymbol{z}(t)\big), \qquad (6)$$

where the iterate $\boldsymbol{z}(t)$ is now a continuous function of time $t$. The solution of this dynamical system is uniquely defined by the choice of the initial condition $\boldsymbol{z}(0) = \boldsymbol{z}_0$.

### 3.1 CONTINUOUS-TIME META-LEARNING

In gradient-based meta-learning, the task-specific adaptation with gradient descent may also be replaced by a gradient vector field in the limit of infinitely small steps. Inspired by prior work in meta-learning (Raghu et al., 2019; Javed & White, 2019), we assume that an embedding network $f_\Phi$ with meta-parameters $\Phi$ is shared across tasks, and only the parameters $\boldsymbol{W}$ of a task-specific linear classifier are adapted, starting at some initialization $\boldsymbol{W}_0$. Instead of being the result of a few steps of gradient descent though, the final parameters $\boldsymbol{W}(T)$ now correspond to integrating an ODE similar to (6) up to a certain horizon $T$, with the initial conditions $\boldsymbol{W}(0) = \boldsymbol{W}_0$. We call this new meta-learning algorithm $\underline{\text{C}}$ontinuous-$\underline{\text{T}}$ime $\underline{\text{M}}$eta-$\underline{\text{L}}$earning[1] (COMLN).

Treating the learning algorithm as a continuous-time process has the notable advantage that the adapted parameter $\boldsymbol{W}(T)$ is now differentiable wrt. the time horizon $T$ (Wiggins, 2003, Chap. 7), in addition to being differentiable wrt. the initial conditions $\boldsymbol{W}_0$—which plays a central role in gradient-based meta-learning, as described in Section 2.1. This allows us to view $T$ as a meta-parameter on par with $\Phi$ and $\boldsymbol{W}_0$, and to effectively optimize the amount of adaptation using stochastic gradient descent (SGD). The meta-learning objective of COMLN can be written as

$$\min_{\Phi, \boldsymbol{W}_0, T} \ \mathbb{E}_\tau \big[ \mathcal{L}\big(\boldsymbol{W}_\tau(T); f_\Phi(\mathcal{D}_\tau^{\text{test}})\big) \big] \qquad (7)$$

$$\text{s.t.} \ \frac{d\boldsymbol{W}_\tau}{dt} = -\nabla \mathcal{L}\big(\boldsymbol{W}_\tau(t); f_\Phi(\mathcal{D}_\tau^{\text{train}})\big) \qquad \boldsymbol{W}_\tau(0) = \boldsymbol{W}_0 \qquad \forall \tau \sim p(\tau), \quad (8)$$

where $f_\Phi(\mathcal{D}_\tau^{\text{train}}) = \{(f_\Phi(\boldsymbol{x}_m), \boldsymbol{y}_m) \mid (\boldsymbol{x}_m, \boldsymbol{y}_m) \in \mathcal{D}_\tau^{\text{train}}\}$ is the embedded training dataset, and $f_\Phi(\mathcal{D}_\tau^{\text{test}})$ is defined similarly for $\mathcal{D}_\tau^{\text{test}}$. In practice, adaptation is implemented using a numerical integration scheme based on an iterative discretization of the problem, such as Runge-Kutta methods. Although a complete discussion of numerical solvers is outside of the scope of this paper, we recommend (Butcher, 2008) for a comprehensive overview of numerical methods for solving ODEs.

### 3.2 THE CHALLENGES OF OPTIMIZING THE META-LEARNING OBJECTIVE

In order to minimize the meta-learning objective of COMLN, it is common practice to use (stochastic) gradient methods; that requires computing its derivatives wrt. the meta-parameters, which we call *meta-gradients*. Our primary goal is to devise an algorithm whose memory requirements do not scale with the amount of adaptation $T$; this would contrast with standard gradient-based meta-learning methods that backpropagate through a sequence of gradient steps (similar to BPTT), where the

---

[1]COMLN is pronounced *chameleon*.

intermediate parameters are stored during adaptation (i.e. $\boldsymbol{\theta}_t^\tau$ for all $t$ in (2)). Since this objective involves the solution $\boldsymbol{W}(T)$ of an ODE, we can use either the adjoint method, or the forward sensitivity method, in order to compute the derivatives wrt. $\Phi$ and $\boldsymbol{W}_0$ (see Section 2.2).

Although the adjoint method has proven to be an effective strategy for learning Neural ODEs, in practice computing the state $\boldsymbol{W}(t)$ backward in time is numerically unstable when applied to a gradient vector field like the one in (8), even for convex loss functions. Figure 2 shows an example where the trajectory of $\boldsymbol{W}(t)$ recomputed backward in time (in red) diverges significantly from the original trajectory (in green) on a quadratic loss function, even though the two should match exactly in theory since they follow the same dynamics. Intuitively, recomputing $\boldsymbol{W}(t)$ backward in time for a gradient vector field requires doing gradient *ascent* on the loss function, which is prone to compounding numerical errors; this is closely related to the loss of entropy observed by Maclaurin et al. (2015). This divergence makes the backward trajectory of $\boldsymbol{W}(t)$ unreliable to find the adjoint state, ruling out the adjoint sensitivity method for computing the meta-gradients in COMLN.

The forward sensitivity method addresses this shortcoming by avoiding the backward pass altogether. However, it can also be particularly expensive here in terms of memory requirements, since the sensitivity state $\mathcal{S}(t)$ in Section 2.2 now corresponds to Jacobian matrices, such as $d\boldsymbol{W}(t)/d\boldsymbol{W}_0$. As the size $d$ of the feature vectors returned by $f_\Phi$ may be very large, this $Nd \times Nd$ Jacobian matrix would be almost impossible to store in practice; for example in our experiments, it can be as large as $d = 16{,}000$ for a ResNet-12 backbone. In Section 4.1, we will show how to apply forward sensitivity in a memory-efficient way, by leveraging the structure of the loss function. This is achieved by carefully decomposing the Jacobian matrices into smaller pieces that follow specific dynamics. We show in Appendix D that unlike the adjoint method, this process is stable.

### 3.3 Connection with Almost No Inner-Loop (ANIL)

Similarly to ANIL (Raghu et al., 2019), COMLN only adapts the parameters $\boldsymbol{W}$ of the last linear layer of the neural network. There is a deeper connection between both algorithms though: while our description of the adaptation in COMLN (Eq. 8) was independent of the choice of the ODE solver used to find the solution $\boldsymbol{W}(T)$ in practice, if we choose an explicit Euler scheme (Euler, 1913; roughly speaking, discretizing (6) from right to left), then the adaptation of COMLN becomes functionally equivalent to ANIL. However, this equivalence can greatly benefit from the memory-efficient algorithm to compute the meta-gradients described in Section 4, based on the forward sensitivity method. This means that using the methods devised here for COMLN, we can effectively compute the meta-gradients of ANIL with a constant memory cost wrt. the number of gradient steps of adaptation, instead of relying on backpropagation (see also Section 4.2).

### 4 Memory-efficient meta-gradients

For some fixed task $\tau$ and $(\boldsymbol{x}_m, \boldsymbol{y}_m) \in \mathcal{D}_\tau^{\text{train}}$, let $\boldsymbol{\phi}_m = f_\Phi(\boldsymbol{x}_m) \in \mathbb{R}^d$ be the embedding of input $\boldsymbol{x}_m$ through the feature extractor $f_\Phi$. Since we are confronted with a classification problem, the loss function of choice $\mathcal{L}(\boldsymbol{W})$ is typically the cross-entropy loss. Böhning (1992) showed that the gradient of the cross-entropy loss wrt. $\boldsymbol{W}$ can be written as

$$\nabla\mathcal{L}\big(\boldsymbol{W}; f_\Phi(\mathcal{D}_\tau^{\text{train}})\big) = \frac{1}{M}\sum_{m=1}^{M}(\boldsymbol{p}_m - \boldsymbol{y}_m)\boldsymbol{\phi}_m^\top, \tag{9}$$

where $\boldsymbol{p}_m = \text{softmax}(\boldsymbol{W}\boldsymbol{\phi}_m)$ is the vector of probabilities returned by the neural network. The key observation here is that the gradient can be decomposed as a sum of $M$ rank-one matrices, where the feature vectors $\boldsymbol{\phi}_m$ are independent of $\boldsymbol{W}$. Therefore we can fully characterize the gradient of the cross-entropy loss with $M$ vectors $\boldsymbol{p}_m - \boldsymbol{y}_m \in \mathbb{R}^N$, as opposed to the full $N \times d$ matrix. This is particularly useful in the context of few-shot classification, where the number of training examples $M$ is small, and typically significantly smaller than the embedding size $d$.

### 4.1 Decomposition of the meta-gradients

We saw in Section 3.2 that the forward sensitivity method was the only stable option to compute the meta-gradients of COMLN. However, naively applying the forward sensitivity equation would involve

quantities that typically scale with $d^2$, which can be too expensive in practice. Using the structure of (9), the Jacobian matrices appearing in the computation of the meta-gradients for COMLN can be decomposed in such a way that only small quantities will depend on time.

**Meta-gradients wrt. $W_0$**  By the chain rule of derivatives, it is sufficient to compute the Jacobian matrix $dW(T)/dW_0$ in order to obtain the meta-gradient wrt. $W_0$. We show in Appendix B.2 that the sensitivity state $dW(t)/dW_0$ can be decomposed as:

$$\frac{dW(t)}{dW_0} = I - \sum_{i=1}^{M} \sum_{j=1}^{M} B_t[i,j] \otimes \phi_i \phi_j^\top, \tag{10}$$

where $\otimes$ is the Kronecker product, and each $B_t[i,j]$ is an $N \times N$ matrix, solution of the following system of ODEs[2]

$$\frac{dB_t[i,j]}{dt} = \mathbb{1}(i=j) A_i(t) - A_i(t) \sum_{m=1}^{M} (\phi_i^\top \phi_m) B_t[m,j] \qquad B_0[i,j] = 0, \tag{11}$$

and $A_i(t)$, defined in Appendix B.2, is also an $N \times N$ matrix that only depends on $W(t)$ and $\phi_i$. The main consequence of this decomposition is that we can simply integrate the augmented ODE in $\{W(t), B_t[i,j]\}$ up to $T$ to obtain the desired Jacobian matrix, along with the adapted parameters $W(T)$. Furthermore, in contrast to naively applying the forward sensitivity method (see Section 3.2), the $M^2$ matrices $B_t[i,j]$ are significantly smaller than the full Jacobian matrix. In fact, we show in Appendix C that we can compute vector-Jacobian products—required for the chain rule—using only these smaller matrices, and without ever having to explicitly construct the full $Nd \times Nd$ Jacobian matrix $dW(t)/dW_0$ with (10).

**Meta-gradients wrt. $\Phi$**  To backpropagate the error through the embedding network $f_\Phi$, we need to first compute the gradients of the outer-loss wrt. the feature vectors $\phi_m$. Again, by the chain rule, we can get these gradients with the Jacobian matrices $dW(T)/d\phi_m$. Similar to (10), we can show that these Jacobian matrices can be decomposed as:

$$\frac{dW(t)}{d\phi_m} = - \left[ s_m(t) \otimes I + \sum_{i=1}^{M} B_t[i,m] W_0 \otimes \phi_i + \sum_{i=1}^{M} \sum_{j=1}^{M} z_t[i,j,m] \phi_j^\top \otimes \phi_i \right], \tag{12}$$

where $s_m(t)$ and $z_t[i,j,m]$ are vectors of size $N$, that follow some dynamics; the exact form of this system of ODEs, as well as the proof of this decomposition, are given in Appendix B.3. Crucially, the only quantities that depend on time are small objects independent of the embedding size $d$. Following the same strategy as above, we can incorporate these vectors in the augmented ODE, and integrate it to get the necessary Jacobians. Once all the $dW(t)/d\phi_m$ are known, for all the training datapoints, we can apply standard backpropagation through $f_\Phi$ to obtain the meta-gradients wrt. $\Phi$.

**Meta-gradient wrt. $T$**  One of the major novelties of COMLN is the capacity to meta-learn the amount of adaptation using stochastic gradient descent. To compute the meta-gradient wrt. the time horizon $T$, we can directly borrow the results derived by Chen et al. (2018) in the context of Neural ODEs, and apply it to our gradient vector field in (8) responsible for adaptation:

$$\frac{d\mathcal{L}\big(W(T); f_\Phi(\mathcal{D}_\tau^{\text{test}})\big)}{dT} = - \left[ \frac{\partial \mathcal{L}\big(W(T); f_\Phi(\mathcal{D}_\tau^{\text{test}})\big)}{\partial W(T)} \right]^\top \frac{\partial \mathcal{L}\big(W(T); f_\Phi(\mathcal{D}_\tau^{\text{train}})\big)}{\partial W(T)}. \tag{13}$$

The proof is available in Appendix B.4. Interestingly, we find that this involves the alignment between the vectors of partial derivatives of the inner-loss and the outer-loss at $W(T)$, which appeared in various contexts in the meta-learning and the multi-task learning literature (Li et al., 2018; Rothfuss et al., 2019; Yu et al., 2020; Von Oswald et al., 2021).

---

[2] Here we used the notation $B_t[i,j]$ to make the dependence on $t$ explicit, without overloading the notation. A more precise notation would be $B[i,j](t)$.

Table 1: Memory required to compute meta-gradients for different algorithms. Exact: the method returns the exact meta-gradients. Full net.: the whole network is adapted, with a number of meta-parameters $|\boldsymbol{\theta}|$. The requirements for checkpointing are taken from (Shaban et al., 2019). Note that typically $M \ll d$ in few-shot learning.

| Model | Exact | Full net. | Memory |
|---|---|---|---|
| MAML (Finn et al., 2017) | ✓ | ✓ | $\mathcal{O}(|\boldsymbol{\theta}| \cdot T)$ |
| ANIL (Raghu et al., 2019) | ✓ | ✗ | $\mathcal{O}(Nd \cdot T)$ |
| Checkpointing (every $\sqrt{T}$ steps) | ✓ | ✓ | $\mathcal{O}(|\boldsymbol{\theta}| \cdot \sqrt{T})$ |
| iMAML (Rajeswaran et al., 2019) | ✗ | ✓ | $\mathcal{O}(|\boldsymbol{\theta}|)$ |
| Forward sensitivity (naive) | ✓ | ✗ | $\mathcal{O}(N^2 d^2 + MNd^2)$ |
| COMLN | ✓ | ✗ | $\mathcal{O}(M^2 N^2 + M^3 N)$ |

## 4.2 MEMORY EFFICIENCY

Although naively applying the forward sensitivity method would be memory intensive, we have shown in Section 4.1 that the Jacobians can be carefully decomposed into smaller pieces. It turns out that even the parameters $\boldsymbol{W}(t)$ can be expressed using the vectors $\boldsymbol{s}_m(t)$ from the decomposition in (12); see Appendix B.1 for details. As a consequence, to compute the adapted parameters $\boldsymbol{W}(T)$ as well as all the necessary meta-gradients, it is sufficient to integrate a dynamical system in $\{\boldsymbol{B}_t[i,j], \boldsymbol{s}_m(t), \boldsymbol{z}_t[i,j,m]\}$ (see Algorithms 1 & 2 in App. A.1), involving exclusively quantities that are independent of the embedding size $d$. Instead, the size of that system scales with $M$ the total number of training examples, which is typically much smaller than $d$ for few-shot classification.

Table 1 shows a comparison of the memory cost for different algorithms. It is important to note that contrary to other standard gradient-based meta-learning methods, the memory requirements of COMLN do not scale with the amount of adaptation $T$ (i.e. the number of gradient steps in MAML & ANIL), while still returning the exact meta-gradients—unlike iMAML (Rajeswaran et al., 2019), which only returns an approximation of the meta-gradients. We verified empirically this efficiency, both in terms of memory and computation costs, in Section 5.2.

## 5 EXPERIMENTS

For our embedding network $f_\Phi$, we consider two commonly used architectures in meta-learning: Conv-4, a convolutional neural network with 4 convolutional blocks, and ResNet-12, a 12-layer residual network (He et al., 2016). Note that following Lee et al. (2019), ResNet-12 does not include a global pooling layer at the end of the network, leading to feature vectors with embedding dimension $d = 16{,}000$. Additional details about these architectures are given in Appendix E. To compute the adapted parameters and the meta-gradients in COMLN, we integrate the dynamical system described in Section 4.2 with a 4th order Runge-Kutta method with a Dormand Prince adaptive step size (Runge, 1895; Dormand & Prince, 1980); we will come back to the choice of this numerical solver in Section 5.2. Furthermore to ensure that $T > 0$, we parametrized it with an exponential activation.

### 5.1 FEW-SHOT IMAGE CLASSIFICATION

We evaluate COMLN on two standard few-shot image classification benchmarks: the *mini*ImageNet (Vinyals et al., 2016) and the *tiered*ImageNet datasets (Ren et al., 2018), both datasets being derived from ILSVRC-2012 (Russakovsky et al., 2015). The process for creating tasks follows the standard procedure from the few-shot classification literature (Santoro et al., 2016), with distinct classes between the different splits. *mini*Imagenet consists of 100 classes, split into 64 training classes, 16 validation classes, and 20 test classes. *tiered*ImageNet consists of 608 classes grouped into 34 high-level categories from ILSVRC-2012, split into 20 training, 6 validation, and 8 testing categories—corresponding to 351/97/160 classes respectively; Ren et al. (2018) argue that separating data according to high-level categories results in a more difficult and more realistic regime.

Table 2 shows the average accuracies of COMLN compared to various meta-learning methods, be it gradient-based or not. For both backbones, COMLN decisively outperforms all other gradient-based

Table 2: Few-shot classification on *mini*ImageNet & *tiered*ImageNet. The average accuracy (%) on 1,000 held-out meta-test tasks is reported with 95% confidence interval. ✓ denotes gradient-based meta-learning algorithms. ⋆ denotes baseline results we executed using the official implementations.

| Model | | Backbone | *mini*ImageNet 5-way | | *tiered*ImageNet 5-way | |
|---|---|---|---|---|---|---|
| | | | **1-shot** | **5-shot** | **1-shot** | **5-shot** |
| MAML (Finn et al., 2017) | ✓ | Conv-4 | $48.70 \pm 1.84$ | $63.11 \pm 0.92$ | $51.67 \pm 1.81$ | $70.30 \pm 1.75$ |
| ANIL (Raghu et al., 2019) | ✓ | Conv-4 | $46.30 \pm 0.40$ | $61.00 \pm 0.60$ | $49.35 \pm 0.26$ | $65.82 \pm 0.12$ |
| Meta-SGD (Li et al., 2017) | ✓ | Conv-4 | $50.47 \pm 1.87$ | $64.03 \pm 0.94$ | $52.80 \pm 0.44$ | $62.35 \pm 0.26$ |
| CAVIA (Zintgraf et al., 2019) | ✓ | Conv-4 | $51.82 \pm 0.65$ | $65.85 \pm 0.55$ | $52.41 \pm 2.64^\star$ | $67.55 \pm 2.05^\star$ |
| iMAML (Rajeswaran et al., 2019) | ✓ | Conv-4 | $49.30 \pm 1.88$ | $59.77 \pm 0.73^\star$ | $38.54 \pm 1.37^\star$ | $60.24 \pm 0.76^\star$ |
| MetaOptNet-RR (Lee et al., 2019) | | Conv-4 | $\mathbf{53.23 \pm 0.59}$ | $69.51 \pm 0.48$ | $54.63 \pm 0.67$ | $\mathbf{72.11 \pm 0.59}$ |
| MetaOptNet-SVM (Lee et al., 2019) | | Conv-4 | $52.87 \pm 0.57$ | $68.76 \pm 0.48$ | $\mathbf{54.71 \pm 0.67}$ | $71.79 \pm 0.59$ |
| **COMLN (Ours)** | ✓ | Conv-4 | $53.01 \pm 0.62$ | $\mathbf{70.54 \pm 0.54}$ | $54.30 \pm 0.69$ | $71.35 \pm 0.57$ |
| MAML (Finn et al., 2017) | ✓ | ResNet-12 | $49.92 \pm 0.65$ | $63.93 \pm 0.59$ | $55.37 \pm 0.74$ | $72.93 \pm 0.60$ |
| ANIL (Raghu et al., 2019) | ✓ | ResNet-12 | $49.65 \pm 0.65$ | $59.51 \pm 0.56$ | $54.77 \pm 0.76$ | $69.28 \pm 0.67$ |
| MetaOptNet-RR (Lee et al., 2019) | | ResNet-12 | $61.41 \pm 0.61$ | $77.88 \pm 0.46$ | $65.36 \pm 0.71$ | $81.34 \pm 0.52$ |
| MetaOptNet-SVM (Lee et al., 2019) | | ResNet-12 | $\mathbf{62.64 \pm 0.61}$ | $\mathbf{78.63 \pm 0.46}$ | $\mathbf{65.99 \pm 0.72}$ | $\mathbf{81.56 \pm 0.53}$ |
| **COMLN (Ours)** | ✓ | ResNet-12 | $59.26 \pm 0.65$ | $77.26 \pm 0.49$ | $62.93 \pm 0.71$ | $81.13 \pm 0.53$ |

meta-learning methods. Compared to methods that explicitly backpropagate through the learning process, such as MAML or ANIL, the performance gain shown by COMLN could be credited to the longer adaptation $T$ it learns, as opposed to a small number of gradient steps—usually about 10 steps; this was fully enabled by our memory-efficient method to compute meta-gradients, which does not scale with the length of adaptation anymore (see Section 4.2). We analyse the evolution of $T$ during meta-training for these different settings in Appendix E.3. In almost all settings, COMLN is even closing the gap with a strong non-gradient-based method like MetaOptNet; the remainder may be explained in part by the training choices made by Lee et al. (2019) (see Appendix E for details).

## 5.2 EMPIRICAL EFFICIENCY OF COMLN

In Section 4.2, we showed that our algorithm to compute the meta-gradients, based on forward differentiation, had a memory cost independent of the length of adaptation $T$. We verify this empirically in Figure 3, where we compare the memory required by COMLN and other methods to compute the meta-gradients on a single task, with a Conv-4 backbone (Figure 4 in Appendix E.2 shows similar results for ResNet-12). To ensure an aligned comparison between discrete and continuous time, we use a conversion corresponding to a learning rate $\alpha = 0.01$ in (2); see Appendix E.2 for a justification. As expected, the memory cost increases for both MAML and ANIL as the number of gradient steps increases, while it remains constant for iMAML and COMLN. Interestingly, we observe that the cost of COMLN is equivalent to the cost of running ANIL for a small number of steps, showing that the additional cost of integrating the augmented ODE in Section 4.2 to compute the meta-gradients is minimal.

Increasing the length of adaptation also has an impact on the time it takes to compute the adapted parameters, and the meta-gradients. Figure 3 (right) shows how the runtime increases with the amount of adaptation for different algorithms. We see that the efficiency of COMLN depends on the numerical solver used. When we use a simple explicit-Euler scheme, the time taken to compute the meta-gradients matches the one of ANIL; this behavior empirically confirms our observation in Section 3.3. When we use an adaptive numerical solver, such as Runge-Kutta (RK) with a Dormand Prince step size, this computation can be significantly accelerated, thanks to the smaller number of function evaluations. In practice, we show in Appendix E.1 that the choice of the ODE solver has a very minimal impact on the accuracy.

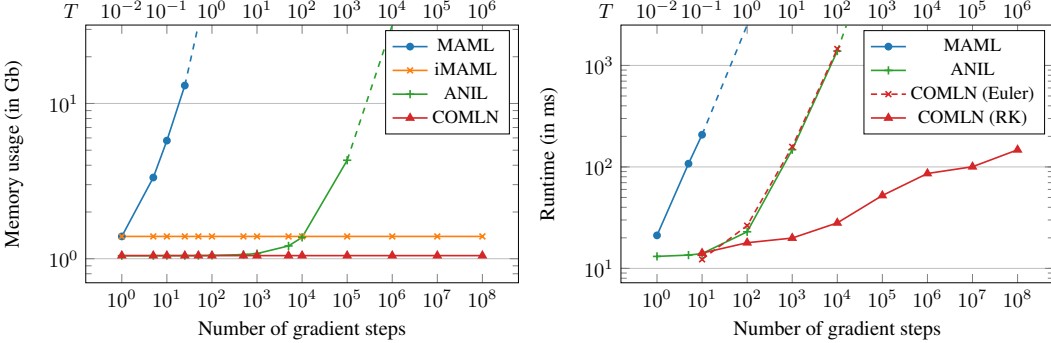

Figure 3: Empirical efficiency of COMLN on a single 5-shot 5-way task, with a Conv-4 backbone. (Left) Memory usage for computing the meta-gradients as a function of the number of inner-gradient steps. The extrapolated dashed lines correspond to the method reaching the memory capacity of a Tesla V100 GPU with 32Gb of memory. (Right) Average time taken (in ms) to compute the exact meta-gradients. The extrapolated dashed lines correspond to the method taking over 2 seconds.

## 6 RELATED WORK

We are interested in meta-learning (Bengio et al., 1991; Schmidhuber, 1987; Thrun & Pratt, 2012), and in particular we focus on gradient-based meta-learning methods (Finn, 2018), where the learning rule is based on gradient descent. While in MAML (Finn et al., 2017) the whole network was updated during this process, follow-up works have shown that it is generally sufficient to share most parts of the neural network, and to only adapt a few layers (Raghu et al., 2019; Chen et al., 2020b; Tian et al., 2020). Even though this hypothesis has been challenged recently (Arnold & Sha, 2021), COMLN also updates only the last layer of a neural network, and therefore can be viewed as a continuous-time extension of ANIL (Raghu et al., 2019); see also Section 3.3. With its shared embedding network across tasks, COMLN is also connected to metric-based meta-learning methods (Vinyals et al., 2016; Snell et al., 2017; Sung et al., 2018; Bertinetto et al., 2018; Lee et al., 2019).

Zhang et al. (2021) introduced a formulation where the adaptation of prototypes follows a gradient vector field, but finally opted for modeling it as a Neural ODE (Chen et al., 2018). Concurrent to our work, Li et al. (2021) also propose a formulation of adaptation based on a gradient vector field, and use the adjoint method to compute the meta-gradients, despite the challenges we identified in Sec. 3.2; Li et al. (2021) also acknowledge these challenges, and they limit their analysis to relatively small values of $T$ (in comparison to the ones learned by COMLN), hence further from the equilibrium, to circumvent this issue altogether. Zhou et al. (2021) also uses a gradient vector field to motivate a novel method with a closed-form adaptation; COMLN still explicitly updates the parameters following the gradient vector field, since there is no closed-form solution of (8). As mentioned in Section 3, treating optimization as a continuous-time process has been used to analyze the convergence of different optimization algorithms, including the meta-optimization of MAML (Xu et al., 2021), or to introduce new meta-optimizers based on different integration schemes (Im et al., 2019). Guo et al. (2021) also uses meta-learning to learn new integration schemes for ODEs. Although this is a growing literature at the intersection of meta-learning and dynamical systems, our work is the first algorithm that uses a gradient vector field for adaptation in meta-learning (see also Li et al. (2021)).

Beyond the memory efficiency of our method, one of the main benefits of the continuous-time perspective is that COMLN is capable of learning when to stop the adaptation, as opposed to taking a number of gradient steps fixed ahead of time. However unlike Chen et al. (2020a), where the number of gradient steps are optimized (up to a maximal number) with variational methods, we incorporate the amount of adaptation as a (continuous) meta-parameter that can be learned using SGD. To compute the meta-gradients, which is known to be challenging for long sequences in gradient-based meta-learning, we use forward-mode differentiation as an alternative to backpropagation through the learning process, similar to prior work in meta-learning (Franceschi et al., 2017; Jiwoong Im et al., 2021) and hyperparameter optimization over long horizons (Micaelli & Storkey, 2021). This yields the exact meta-gradients in constant memory, without any assumption on the optimality of the inner optimization problem, which is necessary when using the normal equations (Bertinetto et al., 2018), or to apply implicit differentiation (Rajeswaran et al., 2019).

## 7 CONCLUSION AND FUTURE WORK

In this paper, we have introduced *Continuous-Time Meta-Learning* (COMLN), a novel algorithm that treats the adaptation in meta-learning as a continuous-time process, by following the dynamics of a gradient vector field up to a certain time horizon $T$. One of the major novelties of treating adaptation in continuous time is that the amount of adaptation $T$ is now a continuous quantity, that can be viewed as a meta-parameter and can be learned using SGD, alongside the initial conditions and the parameters of the embedding network. In order to learn these meta-parameters, we have also introduced a novel practical algorithm based on forward mode automatic differentiation, capable of efficiently computing the exact meta-gradients using an augmented dynamical system. We have verified empirically that this algorithm was able to compute the meta-gradients in constant memory, making it the first gradient-based meta-learning approach capable of computing the exact meta-gradients with long sequences of adaptation using gradient methods. In practice, we have shown that COMLN significantly outperforms other standard gradient-based meta-learning algorithms.

In addition to having a single meta-parameter $T$ that drives the adaptation of all possible tasks, the fact that the time horizon can be learned with SGD opens up new possibilities for gradient-based methods. For example, we could imagine treating $T$ not as a shared meta-parameters, but as a task-specific parameter. This would allow the learning process to be more *adaptive*, possibly with different behaviors depending on the difficulty of the task. This is left as a future direction of research.

### ACKNOWLEDGEMENTS

The authors are grateful to Samsung Electronics Co., Ldt., CIFAR, and IVADO for their funding and Calcul Québec and Compute Canada for providing us with the computing resources.

## REPRODUCIBILITY STATEMENT

We provide in Appendix A.1 a full description in pseudo-code of the meta-training procedure (Algorithm 1), along with the exact dynamics of the ODE (Algorithm 2) and the projection operations (Algorithms 3 & 4) to avoid explicitly building the Jacobian matrices to compute Jacobian-vector products (see Section 4.1).

We also provide in Appendix A.2 a snippet of code in JAX (Bradbury et al., 2018) to compute the adapted parameters $\boldsymbol{W}(T)$, as well as all the necessary objects $\{\boldsymbol{B}_t[i,j], \boldsymbol{s}_m(t), \boldsymbol{z}_t[i,j,m]\}$ to compute all the meta-gradients (see Section 4.2). We also give in Code Snippet 2 the code to compute the meta-gradients wrt. the initialization $\boldsymbol{W}_0$ and the integration time $T$. Computing the meta-gradients wrt. $\Phi$ involves non-minimal dependencies on Haiku (Hennigan et al., 2020), and therefore is omitted here. The full code is available at `https://github.com/tristandeleu/jax-comln`.

**Data generation & hyperparameters** We used the *mini*ImageNet and *tiered*ImageNet datasets provided by Lee et al. (2019) in order to create the 1-shot 5-way and 5-shot 5-way tasks for both datasets. During evaluation, a fixed set of 1,000 tasks was sampled for each setting; this means that both architectures for COMLN have been evaluated using exactly the same data, to ensure direct comparison across backbones. A full description of all the hyperparameters used in COMLN is given in Appendix E.

**Reproducibility of baseline results** To the best of our ability, we have tried to report baseline results from existing work, to limit as much as possible the bias induced by running our own baseline experiments. The references of those works are given in Table 3. We still had to run CAVIA and iMAML on the remaining settings, since these results have not been reported in the literature. For both methods, we used the data generation described above.

- *CAVIA*: We used the official implementation[3]. We used the hyperparameters reported in (Zintgraf et al., 2019) for *mini*ImageNet, and an architecture with 64 filters.
- *iMAML*: We used the official implementation[4]. We used the hyperparameters reported in (Rajeswaran et al., 2019) for *mini*ImageNet 1-shot 5-way.

Table 3: References for the results provided in Table 2: ◯ (Liu et al., 2019), ◯ (Oh et al., 2021), ◯ (Aimen et al., 2021), ◯ (Arnold et al., 2021), and ◯ are reported in their respective references (under *Model*). Recall that $\star$ denotes baseline results we executed using the official implementations.

| Model | | Backbone | *mini*ImageNet 5-way | | *tiered*ImageNet 5-way | |
|---|---|---|---|---|---|---|
| | | | **1-shot** | **5-shot** | **1-shot** | **5-shot** |
| MAML (Finn et al., 2017) | ✓ | Conv-4 | $48.70 \pm 1.84$ | $63.11 \pm 0.92$ | $51.67 \pm 1.81$ | $70.30 \pm 1.75$ |
| ANIL (Raghu et al., 2019) | ✓ | Conv-4 | $46.30 \pm 0.40$ | $61.00 \pm 0.60$ | $49.35 \pm 0.26$ | $65.82 \pm 0.12$ |
| Meta-SGD (Li et al., 2017) | ✓ | Conv-4 | $50.47 \pm 1.87$ | $64.03 \pm 0.94$ | $52.80 \pm 0.44$ | $62.35 \pm 0.26$ |
| CAVIA (Zintgraf et al., 2019) | ✓ | Conv-4 | $51.82 \pm 0.65$ | $65.85 \pm 0.55$ | $52.41 \pm 2.64^\star$ | $67.55 \pm 2.05^\star$ |
| iMAML (Rajeswaran et al., 2019) | ✓ | Conv-4 | $49.30 \pm 1.88$ | $59.77 \pm 0.73^\star$ | $38.54 \pm 1.37^\star$ | $60.24 \pm 0.76^\star$ |
| MetaOptNet-RR (Lee et al., 2019) | | Conv-4 | $\mathbf{53.23 \pm 0.59}$ | $69.51 \pm 0.48$ | $54.63 \pm 0.67$ | $\mathbf{72.11 \pm 0.59}$ |
| MetaOptNet-SVM (Lee et al., 2019) | | Conv-4 | $52.87 \pm 0.57$ | $68.76 \pm 0.48$ | $\mathbf{54.71 \pm 0.67}$ | $71.79 \pm 0.59$ |
| **COMLN (Ours)** | ✓ | Conv-4 | $53.01 \pm 0.62$ | $\mathbf{70.54 \pm 0.54}$ | $54.30 \pm 0.69$ | $71.35 \pm 0.57$ |
| MAML (Finn et al., 2017) | ✓ | ResNet-12 | $49.92 \pm 0.65$ | $63.93 \pm 0.59$ | $55.37 \pm 0.74$ | $72.93 \pm 0.60$ |
| ANIL (Raghu et al., 2019) | ✓ | ResNet-12 | $49.65 \pm 0.65$ | $59.51 \pm 0.56$ | $54.77 \pm 0.76$ | $69.28 \pm 0.67$ |
| MetaOptNet-RR (Lee et al., 2019) | | ResNet-12 | $61.41 \pm 0.61$ | $77.88 \pm 0.46$ | $65.36 \pm 0.71$ | $81.34 \pm 0.52$ |
| MetaOptNet-SVM (Lee et al., 2019) | | ResNet-12 | $\mathbf{62.64 \pm 0.61}$ | $\mathbf{78.63 \pm 0.46}$ | $\mathbf{65.99 \pm 0.72}$ | $\mathbf{81.56 \pm 0.53}$ |
| **COMLN (Ours)** | ✓ | ResNet-12 | $59.26 \pm 0.65$ | $77.26 \pm 0.49$ | $62.93 \pm 0.71$ | $81.13 \pm 0.53$ |

---

[3] `https://github.com/lmzintgraf/cavia/`
[4] `https://github.com/aravindr93/imaml_dev`

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

# Appendix

The Appendix is organized as follows: in Appendix A we give the pseudo-code for meta-training and meta-testing COMLN, along with minimal code in JAX (Bradbury et al., 2018). In Appendix B we prove the decomposition of the Jacobians introduced in Section 4, and we give the exact dynamics of $\boldsymbol{s}_m(t)$ and $\boldsymbol{z}_t[i, j, m]$ in the decomposition of $d\boldsymbol{W}(t)/d\boldsymbol{\phi}_m$, omitted from the main text for concision. In Appendix C we show how the total derivatives may be computed from the Jacobian matrices without ever having to form them explicitly, hence maintaining the memory-efficiency described in Section 4.2. In Appendix D, we show that unlike the adjoint method described in Section 2.2, the algorithm used in COMLN to compute the meta-gradients based on the forward sensitivity method is stable and guaranteed not to diverge. Finally in Appendix E we give additional details regarding the experiments reported in Section 5, together with additional analyses and results on the dataset introduced in Rusu et al. (2018).

## A  ALGORITHMIC DETAILS

### A.1  META-TRAINING PSEUDO-CODE

We give in Algorithm 1 the pseudo-code for meta-training COMLN, based on a distribution of tasks $p(\tau)$, with references to the relevant propositions developed in Appendices B and C. Note that to simplify the presentation in Algorithm 1, we introduce the notations $\nabla_\tau$ and $\partial\mathcal{L}_\tau$ to denote the total derivative of the outer-loss, and the partial derivative of the loss $\mathcal{L}$ respectively, both computed at the adapted parameters $\boldsymbol{W}_\tau(T)$. For example:

$$\nabla_\tau \boldsymbol{W}_0 = \frac{d\mathcal{L}\big(\boldsymbol{W}_\tau(T); f_\Phi(\mathcal{D}_\tau^{\mathrm{test}})\big)}{d\boldsymbol{W}_0} \qquad \partial_{\boldsymbol{W}(T)}\mathcal{L}_\tau^{\mathrm{train}} = \frac{\partial\mathcal{L}\big(\boldsymbol{W}_\tau(T); f_\Phi(\mathcal{D}_\tau^{\mathrm{train}})\big)}{\partial\boldsymbol{W}(T)}$$

---

**Algorithm 1** COMLN – Meta-training

---

**Require:** A task distribution $p(\tau)$
  Initialize randomly $\Phi$ and $\boldsymbol{W}_0$. Initialize $T$ to a small value $\varepsilon > 0$.
  **loop**
    Sample a batch of tasks $\mathcal{B} \sim p(\tau)$
    **for all** $\tau \in \mathcal{B}$ **do**
      Embed the training set: $f_\Phi(\mathcal{D}_\tau^{\mathrm{train}}) = \{(\boldsymbol{\phi}_m, \boldsymbol{y}_m)\}_{m=1}^M$
      $\boldsymbol{s}(T), \boldsymbol{B}_T, \boldsymbol{z}_T \leftarrow \text{ODESolve}([\boldsymbol{0}, \boldsymbol{0}, \boldsymbol{0}], \text{DYNAMICS}, 0, T)$      ▷ Algorithm 2
      $\boldsymbol{W}_\tau(T) \leftarrow \boldsymbol{W}_0 - \sum_m \boldsymbol{s}_m(T)\boldsymbol{\phi}_m^\top$      ▷ Proposition 1

      Embed the test set: $f_\Phi(\mathcal{D}_\tau^{\mathrm{test}})$
      Compute the partial derivatives: $\partial_{\boldsymbol{W}(T)}\mathcal{L}_\tau^{\mathrm{train}}$ & $\partial_{\boldsymbol{W}(T)}\mathcal{L}_\tau^{\mathrm{test}}$
      $\nabla_\tau \boldsymbol{W}_0 \leftarrow \text{PROJECT}_{\boldsymbol{W}_0}\big(\partial_{\boldsymbol{W}(T)}\mathcal{L}_\tau^{\mathrm{test}}, \boldsymbol{B}_T, \boldsymbol{\phi}\big)$      ▷ Proposition 5
      $\nabla_\tau \boldsymbol{\phi}_m \leftarrow \text{PROJECT}_{\boldsymbol{\phi}_m}\big(\partial_{\boldsymbol{W}(T)}\mathcal{L}_\tau^{\mathrm{test}}, \boldsymbol{s}(T), \boldsymbol{B}_T, \boldsymbol{z}_T, \boldsymbol{\phi}\big) + \partial_{\boldsymbol{\phi}_m}\mathcal{L}_\tau^{\mathrm{test}}$      ▷ Proposition 6
      $\nabla_\tau \Phi \leftarrow \text{Backpropagation through } f_\Phi \text{, starting with } \nabla_\tau \boldsymbol{\phi}_m \text{ for all } m$
      $\nabla_\tau T \leftarrow \big(\partial_{\boldsymbol{W}(T)}\mathcal{L}_\tau^{\mathrm{test}}\big)^\top \big(\partial_{\boldsymbol{W}(T)}\mathcal{L}_\tau^{\mathrm{train}}\big)$      ▷ Proposition 4
    **end for**
    Update the meta-parameters:
      $\boldsymbol{W}_0 \leftarrow \boldsymbol{W}_0 - \frac{\alpha}{|\mathcal{B}|} \sum_{\tau \in \mathcal{B}} \nabla_\tau \boldsymbol{W}_0$
      $\Phi \leftarrow \Phi - \frac{\alpha}{|\mathcal{B}|} \sum_{\tau \in \mathcal{B}} \nabla_\tau \Phi$
      $T \leftarrow T - \frac{\alpha}{|\mathcal{B}|} \sum_{\tau \in \mathcal{B}} \nabla_\tau T$
  **end loop**

---

The dynamical system followed during adaptation is given in Propositions 1 to 3 and is summarized in Algorithm 2. The solution of this dynamical system is used not only to find the adapted parameters $\boldsymbol{W}_\tau(T)$, but also to get the quantities necessary to compute the meta-gradients for the updates of

the meta-parameters $\boldsymbol{W}_0$ & $\Phi$. The integration of this dynamical system, named ODESOLVE in Algorithm 1, is done in practice using a numerical solver such as Runge-Kutta methods.

---

**Algorithm 2** Dynamical system

$\quad$**function** DYNAMICS($\boldsymbol{W}_0, f_\Phi(\mathcal{D}_\tau^{\text{train}}), \boldsymbol{s}(t)$)

$\qquad \boldsymbol{W}_\tau(t) \leftarrow \boldsymbol{W}_0 - \sum_m \boldsymbol{s}_m(t)\boldsymbol{\phi}_m^\top$ $\hspace{4cm}$ ▷ Proposition 1

$\qquad \boldsymbol{p}_m(t) \leftarrow \text{softmax}(\boldsymbol{W}_\tau(t)\boldsymbol{\phi}_m)$

$\qquad \boldsymbol{A}_m(t) \leftarrow \big(\text{diag}(\boldsymbol{p}_m(t)) - \boldsymbol{p}_m(t)\boldsymbol{p}_m(t)^\top\big)/M$

$\qquad \dfrac{d\boldsymbol{s}_m}{dt} \leftarrow \dfrac{1}{M}\big(\boldsymbol{p}_m(t) - \boldsymbol{y}_m\big)$ $\hspace{4cm}$ ▷ Proposition 1

$\qquad \dfrac{d\boldsymbol{B}_t[i,j]}{dt} \leftarrow \mathbb{1}(i=j)\boldsymbol{A}_i(t) - \boldsymbol{A}_i(t)\displaystyle\sum_{m=1}^M \big(\boldsymbol{\phi}_i^\top\boldsymbol{\phi}_m\big)\boldsymbol{B}_t[m,j]$ $\hspace{1cm}$ ▷ Propositions 2 & 3 ($\leftrightarrow$)

$\qquad \dfrac{d\boldsymbol{z}_t[i,j,m]}{dt} \leftarrow -\boldsymbol{A}_i(t)\left[\mathbb{1}(i=j)\boldsymbol{s}_m(t) + \mathbb{1}(i=m)\boldsymbol{s}_j(t) + \displaystyle\sum_{k=1}^M \big(\boldsymbol{\phi}_i^\top\boldsymbol{\phi}_k\big)\boldsymbol{z}_t[k,j,m]\right]$

$\qquad$**return** $d\boldsymbol{s}/dt, d\boldsymbol{B}_t/dt, d\boldsymbol{z}_t/dt$

$\quad$**end function**

---

In Algorithms 3 & 4, we give the procedures responsible for the projection of the partial derivatives $\partial_{\boldsymbol{W}(T)}\mathcal{L}_\tau^{\text{test}}$ onto the Jacobian matrices $d\boldsymbol{W}(T)/d\boldsymbol{W}_0$ and $d\boldsymbol{W}(T)/d\boldsymbol{\phi}_m$ respectively, based on Propositions 5 & 6. Note that Algorithm 4 also depends on the initial conditions $\boldsymbol{W}_0$, which is implicitly assumed here for clarity of presentation. It is interesting to see that both functions use the same matrix $\boldsymbol{C}$, and therefore this can be computed only once and reused for both projections.

---

**Algorithm 3** Projection onto $d\boldsymbol{W}(T)/d\boldsymbol{W}_0$

$\quad$**function** PROJECT$_{\boldsymbol{W}_0}(\boldsymbol{V}(T), \boldsymbol{B}_T, \boldsymbol{\phi})$

$\qquad \boldsymbol{C}_j \leftarrow \displaystyle\sum_{i=1}^M \boldsymbol{\phi}_i^\top\boldsymbol{V}(T)^\top\boldsymbol{B}_T[i,j]$

$\qquad$**return** $\boldsymbol{V}(T) - \boldsymbol{C}^\top\boldsymbol{\phi}$

$\quad$**end function**

**Algorithm 4** Projection onto $d\boldsymbol{W}(T)/d\boldsymbol{\phi}_m$

$\quad$**function** PROJECT$_{\boldsymbol{\phi}_m}(\boldsymbol{V}(T), \boldsymbol{s}(T), \boldsymbol{B}_T, \boldsymbol{z}_T, \boldsymbol{\phi})$

$\qquad \boldsymbol{C}_j \leftarrow \displaystyle\sum_{i=1}^M \boldsymbol{\phi}_i^\top\boldsymbol{V}(T)^\top\boldsymbol{B}_T[i,j]$

$\qquad \boldsymbol{D}_{m,j} \leftarrow \displaystyle\sum_{i=1}^M \boldsymbol{z}_T[i,j,m]^\top\boldsymbol{V}(T)\boldsymbol{\phi}_i$

$\qquad$**return** $-\big[\boldsymbol{s}_m(T)^\top\boldsymbol{V}(T) + \boldsymbol{C}_m\boldsymbol{W}_0 + \boldsymbol{D}_m\boldsymbol{\phi}\big]$

$\quad$**end function**

---

Finally in Algorithms 5 & 6, we show how to use COMLN at meta-test time on a novel task, based on the learned meta-parameters $\boldsymbol{W}_0$, $\Phi$ and $T$. This simply corresponds to integrating a dynamical system in $\boldsymbol{W}(t)$ (equivalently, in $\boldsymbol{s}_m(t)$, see Proposition 1). Note that for adaptation during meta-testing, there is no need to compute either $\boldsymbol{B}_t[i,j]$ or $\boldsymbol{z}_t[i,j,m]$, since these are only necessary to compute the meta-gradients during meta-training.

---

**Algorithm 5** COMLN – Meta-test

**Require:** A task $\tau$ with a dataset $\mathcal{D}_\tau^{\text{train}}$

**Require:** Meta-parameters $\Phi$, $\boldsymbol{W}_0$ & $T$

$\quad$Embed the training dataset: $f_\Phi(\mathcal{D}_\tau^{\text{train}})$

$\quad \boldsymbol{s}(T) \leftarrow \text{ODESolve}(\boldsymbol{0}, \text{DYNAMICS}, 0, T)$

$\quad \boldsymbol{W}_\tau(T) \leftarrow \boldsymbol{W}_0 - \sum_m \boldsymbol{s}_m(T)\boldsymbol{\phi}_m^\top$

$\quad$**return** $\boldsymbol{W}_\tau(T) \circ f_\Phi$

**Algorithm 6** Dynamical system for adaptation

$\quad$**function** DYNAMICS($\boldsymbol{W}_0, f_\Phi(\mathcal{D}_\tau^{\text{train}}), \boldsymbol{s}(t)$)

$\qquad \boldsymbol{W}_\tau(t) \leftarrow \boldsymbol{W}_0 - \sum_m \boldsymbol{s}_m(t)\boldsymbol{\phi}_m^\top$

$\qquad \boldsymbol{p}_m(t) \leftarrow \text{softmax}(\boldsymbol{W}_\tau(t)\boldsymbol{\phi}_m)$

$\qquad d\boldsymbol{s}_m/dt \leftarrow \big(\boldsymbol{p}_m(t) - \boldsymbol{y}_m\big)/M$

$\qquad$**return** $d\boldsymbol{s}/dt$

$\quad$**end function**

---

### A.2 SOURCE CODE

We provide a snippet of code written in JAX (Bradbury et al., 2018) in order to compute the adapted parameters, based on Algorithms 1 & 2. The `dynamics` function not only computes the vectors $s_m(t)$ necessary to compute $W(t)$ (see Appendix B.1), but also $B_t[i, j]$ and $z_t[i, j, m]$ jointly in order to compute the meta-gradients. This snippet shows that various necessary quantities can be precomputed ahead of adaptation, such as the Gram matrix `gram`.

```python
import jax.numpy as jnp

from jax import vmap, grad, nn, ops, tree_util
from jax.experimental import ode
from collections import namedtuple

State = namedtuple('State', ['s', 'B', 'z'])

M, N = train_inputs.shape[0], train_labels.shape[1]
gram = jnp.matmul(train_inputs, train_inputs.T)
logits_0 = jnp.matmul(train_inputs, W_0.T)
diag = jnp.diag_indices(M)

def dynamics(state, _):
    preds = nn.softmax(logits_0 - jnp.matmul(gram, state.s), axis=1)
    A = (vmap(jnp.diag)(preds) - vmap(jnp.outer)(preds, preds)) / M

    # Update of s
    ds = (preds - train_labels) / M

    # Update of B
    cross_prod = jnp.einsum('ikn,im,mjnl->ijkl', A, gram, state.B)
    dB = ops.index_add(-cross_prod, diag, A)

    # Update of z
    cross_prod = jnp.einsum('iln,ik,jmn->ijml', A, gram, state.z)
    A_s = jnp.einsum('ikl,jl->ijk', A, state.s)
    dz = ops.index_add(cross_prod, diag, A_s)
    dz = ops.index_add(dz, (diag[0], None, diag[1]), A_s)

    return State(s=ds, B=dB, z=-dz)

state_0 = State(
    s=jnp.zeros((M, N)),
    B=jnp.zeros((M, M, N, N)),
    z=jnp.zeros((M, M, M, N))
)
solution = ode.odeint(dynamics, state_0, jnp.array([0., T]))
state_T = tree_util.tree_map(lambda x: x[-1], solution)
W_T = W_0 - jnp.matmul(state_T.s.T, train_inputs)
```

Code Snippet 1: Snippet of code in JAX to compute the adapted parameter `W_T` for a given task specified by the embedded training set `train_inputs` & `train_labels` (note that here `train_inputs` are the embedding vectors returned by $f_\Phi$) and an initialization `W_0`.

We want to emphasize that all the information required to compute the meta-gradients are available in `state_T` found after integration of the `dynamics` function forward in time, and does not require any additional backward pass (apart from backpropagation through the embedding network $f_\Phi$). In particular, the meta-gradients wrt. the initial conditions $W_0$ and the wrt. the time horizon $T$ can be computed using the snippet of code available in Code Snippet 2.

```
41  grads_test = grad(cross_entropy_loss)(W_T, test_inputs, test_labels)
42  grads_train = grad(cross_entropy_loss)(W_T, train_inputs, train_labels)
43
44  C = jnp.einsum('in,kn,ijkl->jl', train_inputs, grads_test, state_T.B)
45
46  grads_W_0 = grads_test - jnp.matmul(C.T, train_inputs)
47  grads_T = -jnp.vdot(grads_train, grads_test)
```

Code Snippet 2: Snippet of code in JAX to compute the meta-gradients wrt. the meta-parameters W_0 and T, based on `state_T` found above. See Appendix B.4 & Appendix C for details.

The code to compute the meta-gradients wrt. the meta-parameters of the embedding network $\Phi$ is not included here for clarity, as it involves non-minimal dependencies on Haiku (Hennigan et al., 2020) in order to backpropagate the error through the backbone network. The full code is available at https://github.com/tristandeleu/jax-comln.

## B  PROOFS OF MEMORY EFFICIENT META-GRADIENTS

In this section, we will prove the results presented in Section 4 in a slightly more general case, where the loss function $\mathcal{L}$ is the cross-entropy loss regularized with a proximal term around the initialization (Rajeswaran et al., 2019):

$$\mathcal{L}\big(\boldsymbol{W}; f_\Phi(\mathcal{D}_\tau^{\mathrm{train}})\big) = -\frac{1}{M}\sum_{m=1}^{M} \boldsymbol{y}_m^\top \log \boldsymbol{p}_m + \frac{\lambda}{2}\|\boldsymbol{W} - \boldsymbol{W}_0\|_F^2, \tag{14}$$

where $\boldsymbol{p}_m = \mathrm{softmax}(\boldsymbol{W}\boldsymbol{\phi}_m)$, $\|\cdot\|_F$ is the Frobenius norm, and $\lambda$ is a regularization constant. Note that we can recover the setting presented in the main paper by setting $\lambda = 0$. The core idea of computing the meta-gradient efficiently is based on the decomposition of the gradient (as well as the Hessian matrix) of the cross-entropy loss (Böhning, 1992). We recall this decomposition as the following lemma, extended to include the regularization term:

**Lemma 1** (Böhning, 1992). *Let $f_\Phi(\mathcal{D}_\tau^{\mathrm{train}}) = \{(\boldsymbol{\phi}_m, \boldsymbol{y}_m)\}_{m=1}^M$ be the embedded training set through the embedding network $f_\Phi$, and $\mathcal{L}$ the regularized cross-entropy loss defined in (14). The gradient $\nabla\mathcal{L}$ and the Hessian matrix $\nabla^2\mathcal{L}$ of the regularized cross-entropy loss can be written as*

$$\nabla\mathcal{L}\big(\boldsymbol{W}; f_\Phi(\mathcal{D}_\tau^{\mathrm{train}})\big) = \frac{1}{M}\sum_{m=1}^{M}\big(\boldsymbol{p}_m - \boldsymbol{y}_m\big)\boldsymbol{\phi}_m^\top + \lambda(\boldsymbol{W} - \boldsymbol{W}_0)$$

$$\nabla^2\mathcal{L}\big(\boldsymbol{W}; f_\Phi(\mathcal{D}_\tau^{\mathrm{train}})\big) = \sum_{m=1}^{M}\boldsymbol{A}_m \otimes \boldsymbol{\phi}_m\boldsymbol{\phi}_m^\top + \lambda\boldsymbol{I},$$

*where $\boldsymbol{A}_m = \big(\mathrm{diag}(\boldsymbol{p}_m) - \boldsymbol{p}_m\boldsymbol{p}_m^\top\big)/M$ are $N \times N$ matrices, and $\otimes$ is the Kronecker product.*

This lemma is particularly useful in the context of few-shot learning since it reduces the characterization of the gradient and the Hessian from quantities of size $N \times d$ and $Nd \times Nd$ respectively (where $d$ is the dimension of the embedding vectors $\phi$) to $M$ objects whose size is independent of $d$—the embedding vectors $\phi$ being independent of $\boldsymbol{W}$. Typically in few-shot learning, $M \ll d$.

In order to avoid higher-order tensors when defining the different Jacobians and Hessians, we always consider them as matrices, with possibly an implicit "flattening" operation. For example here even though $\boldsymbol{W}$ is a $N \times d$ matrix, we treat the Hessian $\nabla^2\mathcal{L}$ as a $Nd \times Nd$ matrix, as opposed to a 4D tensor. When the context is required, we will make this transformation from a higher-order tensor to a matrix explicit with an encoding of indices. Moreover throughout this section, we will only consider the computation of the meta-gradients for a single task $\tau$, and therefore we will often drop the explicit dependence of the different objects on $\tau$ (e.g. we will write $\boldsymbol{W}(t)$ instead of $\boldsymbol{W}_\tau(t)$); the meta-gradients are eventually averaged over a batch of tasks for the update of the outer-loop, see Appendix A for details in the pseudo-code. Finally, since we only consider adaptation of the last layer of the neural network, the presentation here is always made in the context of an embedded training set $f_\Phi(\mathcal{D}_\tau^{\mathrm{train}}) = \{(\boldsymbol{\phi}_m, \boldsymbol{y}_m)\}_{m=1}^M$ that went through the backbone $f_\Phi$.

### B.1 DECOMPOSITION OF $W(t)$ FOR PARAMETER ADAPTATION

As a direct application of Lemma 1, we first decompose the parameters $W(t)$ into smaller quantities $s_m(t)$ that follow the dynamics defined in Prop. 1. Although this decomposition is equivalent to $W(t)$, in practice solving a smaller dynamical system in $s_m(t)$ improves the efficiency of our method.

**Proposition 1.** *Let* $f_\Phi(\mathcal{D}_\tau^{\text{train}}) = \{(\phi_m, y_m)\}_{m=1}^M$ *be the embedded training set through the embedding network* $f_\Phi$, *and* $W(t)$ *be the solution of the following dynamical system*

$$\frac{dW}{dt} = -\nabla\mathcal{L}\big(W(t); f_\Phi(\mathcal{D}_\tau^{\text{train}})\big) \qquad\qquad W(0) = W_0,$$

*where the loss function is the regularized cross-entropy loss defined in (14). The solution* $W(t)$ *of this dynamical system can be written as*

$$W(t) = W_0 - \sum_{m=1}^M s_m(t)\phi_m^\top,$$

*where for all* $m$, $s_m(t)$ *is the solution of the following dynamical system:*

$$\frac{ds_m}{dt} = \frac{1}{M}\big(p_m(t) - y_m\big) - \lambda s_m(t) \qquad\qquad s_m(0) = 0,$$

*and* $p_m(t) = \text{softmax}(W(t)\phi_m)$ *is the vector of predictions returned by the network at time* $t$.

*Proof.* Using Lemma 1, the function $W(t)$ is the solution of the following differential equation:

$$\begin{aligned}
\frac{dW}{dt} &= -\nabla\mathcal{L}\big(W(t); f_\Phi(\mathcal{D}_\tau^{\text{train}})\big) \\
&= -\frac{1}{M}\sum_{m=1}^M \big(p_m(t) - y_m\big)\phi_m^\top - \lambda\big(W(t) - W_0\big) \qquad W(0) = W_0.
\end{aligned}$$

The proof relies on the unicity of the solution of a given autonomous differential equation given a particular choice of initial conditions (Wiggins, 2003, Prop. 7.4.2). In other words, if we can find another function $\widetilde{W}(t)$ that also satisfies the differential equation above with the initial conditions $\widetilde{W}(0) = W_0$, then it means that for all $t$ we have $\widetilde{W}(t) = W(t)$. Suppose that this function $\widetilde{W}(t)$ can be written as

$$\widetilde{W}(t) = W_0 - \sum_{m=1}^M s_m(t)\phi_m^\top,$$

where $s_m$ satisfies the dynamical system defined in the statement of Proposition 1. Then we have

$$\begin{aligned}
-\nabla\mathcal{L}\big(\widetilde{W}(t); f_\Phi(\mathcal{D}_\tau^{\text{train}})\big) &= -\frac{1}{M}\sum_{m=1}^M \big(p_m(t) - y_m\big)\phi_m^\top - \lambda\big(\widetilde{W}(t) - W_0\big) \\
&= -\sum_{m=1}^M \frac{ds_m}{dt}\phi_m^\top - \underbrace{\lambda\sum_{m=1}^M s_m(t)\phi_m^\top - \lambda\big(\widetilde{W}(t) - W_0\big)}_{=\,0} \\
&= -\sum_{m=1}^M \frac{ds_m}{dt}\phi_m^\top = \frac{d\widetilde{W}}{dt}
\end{aligned}$$

We have shown that $\widetilde{W}(t)$ follows the same dynamics as $W(t)$. Moreover, using the initial conditions $s_m(0) = 0$, it is clear that $\widetilde{W}(0) = W_0$, which are the same initial conditions as the equation satisfied by $W(t)$. Therefore we have $\widetilde{W}(t) = W(t)$ for all $t$, showing the expected decomposition of $W(t)$. $\qquad\square$

### B.2 Decomposition of the Jacobian matrix $d\boldsymbol{W}(t)/d\boldsymbol{W}_0$

The core objective of gradient-based meta-learning methods is the capacity to compute the meta-gradients wrt. the initial conditions $\boldsymbol{W}_0$. In order to compute this meta-gradient using forward-mode automatic differentiation, we want to first compute the Jacobian matrix $d\boldsymbol{W}(t)/d\boldsymbol{W}_0$. We show that this Jacobian can be decomposed into smaller quantities that follow the dynamics in Proposition 2.

**Proposition 2.** *Let* $f_\Phi(\mathcal{D}_\tau^{\mathrm{train}}) = \{(\boldsymbol{\phi}_m, \boldsymbol{y}_m)\}_{m=1}^M$ *be the embedded training set through the embedding network* $f_\Phi$. *The Jacobian matrix* $d\boldsymbol{W}(t)/d\boldsymbol{W}_0$ *can be written as*

$$\frac{d\boldsymbol{W}(t)}{d\boldsymbol{W}_0} = \boldsymbol{I} - \sum_{i=1}^M \sum_{j=1}^M \boldsymbol{B}_t[i,j] \otimes \boldsymbol{\phi}_i \boldsymbol{\phi}_j^\top,$$

*where* $\otimes$ *is the Kronecker product, and for all* $i,j$, $\boldsymbol{B}_t[i,j]$ *is a* $N \times N$ *matrix which the solution of the following dynamical system*

$$\frac{d\boldsymbol{B}_t[i,j]}{dt} = \mathbb{1}(i=j)\boldsymbol{A}_i(t) - \lambda\boldsymbol{B}_t[i,j] - \boldsymbol{A}_i(t)\sum_{m=1}^M \big(\boldsymbol{\phi}_i^\top\boldsymbol{\phi}_m\big)\boldsymbol{B}_t[m,j] \qquad \boldsymbol{B}_0[i,j] = \boldsymbol{0},$$

*and* $\boldsymbol{A}_i(t) = \big(\mathrm{diag}(\boldsymbol{p}_i(t)) - \boldsymbol{p}_i(t)\boldsymbol{p}_i(t)^\top\big)/M$ *are defined using the vectors of predictions at time* $t$ $\boldsymbol{p}_i(t) = \mathrm{softmax}\big(\boldsymbol{W}(t)\boldsymbol{\phi}_i\big)$.

*Proof.* We will use the forward sensitivity equation from Section 2.2 in order to derive this new dynamical system over $\boldsymbol{B}_t[i,j]$. Recall that to simplify the notations we can write the gradient vector field followed during adaptation as

$$\frac{d\boldsymbol{W}}{dt} = g\big(\boldsymbol{W}(t); f_\Phi(\mathcal{D}_\tau^{\mathrm{train}})\big) \triangleq -\nabla\mathcal{L}\big(\boldsymbol{W}(t); f_\Phi(\mathcal{D}_\tau^{\mathrm{train}})\big).$$

Introducing the matrix-valued function $\mathcal{S}(t) = d\boldsymbol{W}(t)/d\boldsymbol{W}_0$ as a sensitivity state, we can use the forward sensitivity equations and see that $\mathcal{S}(t)$ satisfies the following equation

$$\frac{d\mathcal{S}}{dt} = \frac{\partial g\big(\boldsymbol{W}(t)\big)}{\partial\boldsymbol{W}(t)}\mathcal{S}(t) + \frac{\partial g\big(\boldsymbol{W}(t)\big)}{\partial\boldsymbol{W}_0} \qquad\qquad \mathcal{S}(0) = \boldsymbol{I}$$

$$= -\nabla^2\mathcal{L}\big(\boldsymbol{W}(t); f_\Phi(\mathcal{D}_\tau^{\mathrm{train}})\big)\mathcal{S}(t) + \lambda\boldsymbol{I}.$$

The rest of the proof is based on the unicity of the solution of a given autonomous[5] differential equation given a particular choice of initial conditions (Wiggins, 2003, Prop. 7.4.2). In other words, if we can find another function $\widetilde{\mathcal{S}}(t)$ that also satisfies the above differential equation with the initial conditions $\widetilde{\mathcal{S}}(0) = \boldsymbol{I}$, then it means that for all $t$ we have $\widetilde{\mathcal{S}}(t) = \mathcal{S}(t)$. Suppose that this function can be written as

$$\widetilde{\mathcal{S}}(t) = \boldsymbol{I} - \sum_{i=1}^M \sum_{j=1}^M \boldsymbol{B}_t[i,j] \otimes \boldsymbol{\phi}_i\boldsymbol{\phi}_j^\top,$$

where $\boldsymbol{B}_t[i,j]$ satisfies the dynamical system defined in the statement of Proposition 2. Then we have, using Lemma 1:

$$- \nabla^2\mathcal{L}\big(\boldsymbol{W}(t); f_\Phi(\mathcal{D}_\tau^{\mathrm{train}})\big)\widetilde{\mathcal{S}}(t) + \lambda\boldsymbol{I}$$

$$= -\bigg[\sum_{i=1}^M \boldsymbol{A}_i(t) \otimes \boldsymbol{\phi}_i\boldsymbol{\phi}_i^\top + \lambda\boldsymbol{I}\bigg]\bigg[\boldsymbol{I} - \sum_{m,j}\boldsymbol{B}_t[m,j] \otimes \boldsymbol{\phi}_m\boldsymbol{\phi}_j^\top\bigg] + \lambda\boldsymbol{I}$$

$$= -\sum_{i=1}^M \boldsymbol{A}_i(t) \otimes \boldsymbol{\phi}_i\boldsymbol{\phi}_i^\top + \sum_{i,j,m} \big(\boldsymbol{A}_i(t)\boldsymbol{B}_t[m,j]\big) \otimes \big(\boldsymbol{\phi}_i \underbrace{\boldsymbol{\phi}_i^\top\boldsymbol{\phi}_m}_{\in\mathbb{R}} \boldsymbol{\phi}_j^\top\big) + \lambda\sum_{i,j}\boldsymbol{B}_t[i,j] \otimes \boldsymbol{\phi}_i\boldsymbol{\phi}_j^\top$$

---

[5]While the dynamical system defined here for the sensitivity state $\mathcal{S}(t)$ alone is not exactly autonomous due to the dependence of the Hessian matrix on $\boldsymbol{W}(t)$, we could augment $\mathcal{S}(t)$ with $\boldsymbol{W}(t)$ to obtain an autonomous system on the augmented state. We come back to this distinction in Appendix D. The unicity argument still holds here (the augmented solution would be unique).

$$= -\sum_{i,j} \left[ \mathbb{1}(i = j) \boldsymbol{A}_i(t) - \lambda \boldsymbol{B}_t[i, j] - \boldsymbol{A}_i(t) \sum_{m=1}^{M} \left( \boldsymbol{\phi}_i^\top \boldsymbol{\phi}_m \right) \boldsymbol{B}_t[m, j] \right] \otimes \boldsymbol{\phi}_i \boldsymbol{\phi}_j^\top$$

$$= -\sum_{i,j} \frac{d\boldsymbol{B}_t[i, j]}{dt} \otimes \boldsymbol{\phi}_i \boldsymbol{\phi}_j^\top = \frac{d\widetilde{\mathcal{S}}}{dt}$$

We have shown that $\widetilde{\mathcal{S}}(t)$ follows the same dynamics as $\mathcal{S}(t)$. Moreover, using the initial conditions $\boldsymbol{B}_0[i, j] = \boldsymbol{0}$, it is clear that $\widetilde{\mathcal{S}}(0) = \boldsymbol{I}$, which are the same initial conditions as the equation satisfied by $\mathcal{S}(t)$. Therefore, we have $\widetilde{\mathcal{S}}(t) = \mathcal{S}(t)$ for all $t$, showing the expected decomposition of $\mathcal{S}(t) = d\boldsymbol{W}(t)/d\boldsymbol{W}_0$. $\qquad \square$

## B.3 DECOMPOSITION OF THE JACOBIAN MATRIX $d\boldsymbol{W}(t)/d\boldsymbol{\phi}_m$

Similar to Proposition 2, there exists a decomposition of the Jacobian matrix $d\boldsymbol{W}(T)/d\boldsymbol{\phi}_m$. Recall that this Jacobian matrix appears in the computation of the gradient of the outer-loss wrt. the embedding vectors $\boldsymbol{\phi}_m$, which is necessary in order to compute the meta-gradients wrt. the meta-parameters of the embedding network $f_\Phi$ using backpropagation from the last layer of $f_\Phi$.

**Proposition 3.** *Let $f_\Phi(\mathcal{D}_\tau^{\mathrm{train}}) = \{(\boldsymbol{\phi}_m, \boldsymbol{y}_m)\}_{m=1}^{M}$ be the embedded training set through the embedding network $f_\Phi$. For all $m$, the Jacobian matrix $d\boldsymbol{W}(t)/d\boldsymbol{\phi}_m$ can be written as*

$$\frac{d\boldsymbol{W}(t)}{d\boldsymbol{\phi}_m} = -\left[ \boldsymbol{s}_m(t) \otimes \boldsymbol{I}_d + \sum_{i=1}^{M} \boldsymbol{B}_t[i, m] \boldsymbol{W}_0 \otimes \boldsymbol{\phi}_i + \sum_{i=1}^{M} \sum_{j=1}^{M} \boldsymbol{z}_t[i, j, m] \boldsymbol{\phi}_j^\top \otimes \boldsymbol{\phi}_i \right],$$

*where $\boldsymbol{s}_m(t)$ and $\boldsymbol{B}_t[i, j]$ are the solutions of the ODEs defined in Propositions 1 & 2, and for all $i, j, m$, $\boldsymbol{z}_t[i, j, m]$ is a vector of length $N$ solution of the following dynamical system:*

$$\frac{d\boldsymbol{z}_t[i, j, m]}{dt} = -\boldsymbol{A}_i(t) \left[ \mathbb{1}(i = j) \boldsymbol{s}_m(t) + \mathbb{1}(i = m) \boldsymbol{s}_j(t) + \lambda \boldsymbol{z}_t[i, j, m] + \sum_{k=1}^{M} \left( \boldsymbol{\phi}_i^\top \boldsymbol{\phi}_k \right) \boldsymbol{z}_t[k, j, m] \right],$$

*with the initial conditions $\boldsymbol{z}_0[i, j, m] = \boldsymbol{0}$.*

*Proof.* The outline of this proof follows the proof of Proposition 2: we first use the forward sensitivity equations to get the dynamical system followed by the Jacobian matrix $d\boldsymbol{W}(t)/d\boldsymbol{\phi}_m$, and then use a unicity argument given that the new decomposition satisfies the same ODE. Recall that we use the following notation to write the gradient vector field followed during adaptation:

$$\frac{d\boldsymbol{W}}{dt} = g\big(\boldsymbol{W}(t); f_\Phi(\mathcal{D}_\tau^{\mathrm{train}})\big) \triangleq -\nabla \mathcal{L}\big(\boldsymbol{W}(t); f_\Phi(\mathcal{D}_\tau^{\mathrm{train}})\big).$$

For a fixed $m$, we introduce the matrix valued function $\mathcal{S}(t) = d\boldsymbol{W}(t)/d\boldsymbol{\phi}_m$ as a sensitivity state. We can use the forward sensitivity equations, and see that $\mathcal{S}(t)$ satisfies the following equation

$$\frac{d\mathcal{S}}{dt} = \frac{\partial g\big(\boldsymbol{W}(t), \boldsymbol{\phi}_m\big)}{\partial \boldsymbol{W}(t)} \mathcal{S}(t) + \frac{\partial g\big(\boldsymbol{W}(t), \boldsymbol{\phi}_m\big)}{\partial \boldsymbol{\phi}_m} \qquad\qquad \mathcal{S}(0) = \frac{d\boldsymbol{W}(0)}{d\boldsymbol{\phi}_m} = \boldsymbol{0}$$

$$= -\nabla^2 \mathcal{L}\big(\boldsymbol{W}(t); f_\Phi(\mathcal{D}_\tau^{\mathrm{train}})\big) \mathcal{S}(t) - \boldsymbol{A}_m(t) \boldsymbol{W}_0 \otimes \boldsymbol{\phi}_m$$

$$+ \boldsymbol{A}_m(t) \sum_{i=1}^{M} \boldsymbol{s}_i(t) \boldsymbol{\phi}_i^\top \otimes \boldsymbol{\phi}_m - \frac{1}{M} \big(\boldsymbol{p}_m(t) - \boldsymbol{y}_m\big) \otimes \boldsymbol{I}_d$$

where we make the direct dependence of $g$ on $\boldsymbol{\phi}_m$ explicit, and we used the decomposition of $\boldsymbol{W}(t)$ from Proposition 1. Suppose that we define the function $\widetilde{\mathcal{S}}(t)$ as

$$\widetilde{\mathcal{S}}(t) = -\left[ \boldsymbol{s}_m(t) \otimes \boldsymbol{I}_d + \sum_{i=1}^{M} \boldsymbol{B}_t[i, m] \boldsymbol{W}_0 \otimes \boldsymbol{\phi}_i + \sum_{i,j} \boldsymbol{z}_t[i, j, m] \boldsymbol{\phi}_j^\top \otimes \boldsymbol{\phi}_i \right],$$

where $\boldsymbol{z}_t[i,j,m]$ satisfies the dynamical system defined in the statement of Proposition 3. Then we have, using Lemma 1:

$$-\nabla^2\mathcal{L}\big(\boldsymbol{W}(t); f_\Phi(\mathcal{D}_\tau^{\mathrm{train}})\big)\widetilde{\mathcal{S}}(t) - \boldsymbol{A}_m(t)\bigg[\boldsymbol{W}_0 - \sum_{i=1}^M \boldsymbol{s}_i(t)\boldsymbol{\phi}_i^\top\bigg]\otimes\boldsymbol{\phi}_m - \frac{1}{M}\big(\boldsymbol{p}_m(t) - \boldsymbol{y}_m\big)\otimes\boldsymbol{I}_d$$

$$= \bigg[\sum_{i=1}^M \boldsymbol{A}_i(t)\otimes\boldsymbol{\phi}_i\boldsymbol{\phi}_i^\top + \lambda\boldsymbol{I}\bigg]\bigg[\boldsymbol{s}_m(t)\otimes\boldsymbol{I}_d + \sum_{i=1}^M \boldsymbol{B}_t[i,m]\boldsymbol{W}_0\otimes\boldsymbol{\phi}_i + \sum_{i,j}\boldsymbol{z}_t[i,j,m]\boldsymbol{\phi}_j^\top\otimes\boldsymbol{\phi}_i\bigg]$$

$$- \boldsymbol{A}_m(t)\boldsymbol{W}_0\otimes\boldsymbol{\phi}_m + \boldsymbol{A}_m(t)\sum_{i=1}^M \boldsymbol{s}_i(t)\boldsymbol{\phi}_i^\top\otimes\boldsymbol{\phi}_m - \frac{1}{M}\big(\boldsymbol{p}_m(t) - \boldsymbol{y}_m\big)\otimes\boldsymbol{I}_d$$

$$= -\underbrace{\bigg[\frac{1}{M}\big(\boldsymbol{p}_m(t) - \boldsymbol{y}_m\big) - \lambda\boldsymbol{s}_m(t)\bigg]}_{=d\boldsymbol{s}_m/dt}\otimes\boldsymbol{I}_d$$

$$- \sum_{i=1}^M \underbrace{\bigg[\mathbb{1}(i=m)\boldsymbol{A}_i(t) - \lambda\boldsymbol{B}_t[i,m] - \boldsymbol{A}_i(t)\sum_{k=1}^M \big(\boldsymbol{\phi}_i^\top\boldsymbol{\phi}_k\big)\boldsymbol{B}_t[k,m]\bigg]}_{=d\boldsymbol{B}_t[i,m]/dt}\boldsymbol{W}_0\otimes\boldsymbol{\phi}_i$$

$$+ \sum_{i,j}\boldsymbol{A}_i(t)\underbrace{\bigg[\mathbb{1}(i=j)\boldsymbol{s}_m(t) + \mathbb{1}(i=m)\boldsymbol{s}_j(t) + \lambda\boldsymbol{z}_t[i,j,m] + \sum_{k=1}^M \big(\boldsymbol{\phi}_i^\top\boldsymbol{\phi}_k\big)\boldsymbol{z}_t[k,j,m]\bigg]}_{=-d\boldsymbol{z}_t[i,j,m]/dt}\boldsymbol{\phi}_j^\top\otimes\boldsymbol{\phi}_i$$

$$= \frac{d\widetilde{\mathcal{S}}}{dt}$$

We have shown that $\widetilde{\mathcal{S}}(t)$ follows the same dynamics as $\mathcal{S}(t)$. Moreover using the initial conditions $\boldsymbol{s}_m(0) = \boldsymbol{0}$, $\boldsymbol{B}_0[i,j] = \boldsymbol{0}$, and $\boldsymbol{z}_0[i,j,m] = \boldsymbol{0}$, we have $\widetilde{\mathcal{S}}(0) = \boldsymbol{0}$, which are the same initial conditions as the equation satisfied by $\mathcal{S}(t)$. Therefore we have $\widetilde{\mathcal{S}}(t) = \mathcal{S}(t)$ for all $t$, showing the expected decomposition of the Jacobian matrix $\mathcal{S}(t) = d\boldsymbol{W}(t)/d\boldsymbol{\phi}_m$. $\qquad\square$

### B.4 Proof of the meta-gradient wrt. $T$

The novelty of COMLN over prior work on gradient-based meta-learning is the ability to meta-learn the amount of adaptation $T$ using SGD. To compute the meta-gradient wrt. the time horizon $T$, we can apply the result from Chen et al. (2018).

**Proposition 4.** *Let $f_\Phi(\mathcal{D}_\tau^{\mathrm{train}})$ and $f_\Phi(\mathcal{D}_\tau^{\mathrm{test}})$ be the embedding through the network $f_\Phi$ of the training and test set respectively. The gradient of the outer-loss wrt. the time horizon $T$ is given by:*

$$\frac{d\mathcal{L}\big(\boldsymbol{W}(T); f_\Phi(\mathcal{D}_\tau^{\mathrm{test}})\big)}{dT} = -\bigg[\frac{\partial\mathcal{L}\big(\boldsymbol{W}(T); f_\Phi(\mathcal{D}_\tau^{\mathrm{test}})\big)}{\partial\boldsymbol{W}(T)}\bigg]^\top \frac{\partial\mathcal{L}\big(\boldsymbol{W}(T); f_\Phi(\mathcal{D}_\tau^{\mathrm{train}})\big)}{\partial\boldsymbol{W}(T)}.$$

*Proof.* We can directly apply the result from (Chen et al., 2018, App. B.2), which we recall here for completeness. Given the ODE $d\boldsymbol{z}/dt = g(\boldsymbol{z}(t))$, the gradient of the loss $\widetilde{\mathcal{L}}\big(\boldsymbol{z}(T)\big)$ wrt. $T$ is given by

$$\frac{d\widetilde{\mathcal{L}}}{dT} = \boldsymbol{a}(T)^\top g(\boldsymbol{z}(T)) \qquad \text{where} \quad \boldsymbol{a}(T) = \frac{\partial\widetilde{\mathcal{L}}}{\partial\boldsymbol{z}(T)}$$

is the *initial adjoint state* (see Section 2.2). In our case we have

$$g(\boldsymbol{z}(T)) \equiv -\nabla\mathcal{L}\big(\boldsymbol{W}(T); f_\Phi(\mathcal{D}_\tau^{\mathrm{train}})\big) \qquad \boldsymbol{a}(T) \equiv \frac{\partial\mathcal{L}\big(\boldsymbol{W}(T); f_\Phi(\mathcal{D}_\tau^{\mathrm{test}})\big)}{\partial\boldsymbol{W}(T)}.$$

$\qquad\square$

## C  PROJECTION ONTO THE JACOBIAN MATRICES

Once we have computed the Jacobian $d\boldsymbol{W}(T)/d\boldsymbol{\theta}$ wrt. some meta-parameter $\boldsymbol{\theta}$ (in our case, either the initial conditions $\boldsymbol{W}_0$ or the embedding vectors $\boldsymbol{\phi}_m$), we only have to project the vector of partial derivatives onto it to obtain the meta-gradients (vector-Jacobian product), using the chain rule:

$$\frac{d\mathcal{L}\big(\boldsymbol{W}(T); f_\Phi(\mathcal{D}_\tau^{\text{test}})\big)}{d\boldsymbol{\theta}} = \frac{\partial\mathcal{L}\big(\boldsymbol{W}(T); f_\Phi(\mathcal{D}_\tau^{\text{test}})\big)}{\partial\boldsymbol{W}(T)} \frac{d\boldsymbol{W}(T)}{d\boldsymbol{\theta}} + \frac{\partial\mathcal{L}\big(\boldsymbol{W}(T); f_\Phi(\mathcal{D}_\tau^{\text{test}})\big)}{\partial\boldsymbol{\theta}}.$$

While we have shown how to decompose the Jacobian matrices in such a way that it only involves quantities that are independent of $d$ the dimension of the embedding vectors $\boldsymbol{\phi}$, this final projection a priori requires us to form the Jacobian explicitly. Even though this operation is done only once at the end of the integration, this may be overly expensive since the Jacobian matrices scale quadratically with $d$, which can be as high as $d = 16{,}000$ in our experiments.

Fortunately, we can perform this projection as a function of $\boldsymbol{s}_m(T)$, $\boldsymbol{B}_T[i, j]$, and $\boldsymbol{z}_T[i, j, m]$, without having to explicitly form the full Jacobian matrix, by exchanging the order of operations.

**Proposition 5.** *Let $f_\Phi(\mathcal{D}_\tau^{\text{train}}) = \{(\boldsymbol{\phi}_m, \boldsymbol{y}_m)\}_{m=1}^M$ and $f_\Phi(\mathcal{D}_\tau^{\text{test}})$ be the embedded training set and test set through the embedding network $f_\Phi$ respectively. Let $\boldsymbol{B}_T[i, j]$ be the solution at time $T$ of the differential equation defined in [Proposition 2](), and $\boldsymbol{V}(T) \in \mathbb{R}^{N \times d}$ the partial derivative of the outer-loss wrt. the adapted parameters $\boldsymbol{W}(T)$:*

$$\boldsymbol{V}(T) = \frac{\partial\mathcal{L}\big(\boldsymbol{W}(T); f_\Phi(\mathcal{D}_\tau^{\text{test}})\big)}{\partial\boldsymbol{W}(T)} \quad \text{so that} \quad \frac{d\mathcal{L}\big(\boldsymbol{W}(T); f_\Phi(\mathcal{D}_\tau^{\text{test}})\big)}{d\boldsymbol{W}_0} = \text{Vec}\big(\boldsymbol{V}(T)\big)^T \frac{d\boldsymbol{W}(T)}{d\boldsymbol{W}_0}.$$

*Let $\boldsymbol{\phi} = [\boldsymbol{\phi}_1, \dots, \boldsymbol{\phi}_M]^\top \in \mathbb{R}^{M \times d}$ be the design matrix. The gradient of the outer-loss wrt. the initial conditions $\boldsymbol{W}_0$ can be computed as*

$$\frac{d\mathcal{L}\big(\boldsymbol{W}(T); f_\Phi(\mathcal{D}_\tau^{\text{test}})\big)}{d\boldsymbol{W}_0} = \boldsymbol{V}(T) - \boldsymbol{C}^\top \boldsymbol{\phi},$$

*where the rows of $\boldsymbol{C} \in \mathbb{R}^{M \times N}$ are defined by*

$$\boldsymbol{C}_j = \sum_{i=1}^M \boldsymbol{\phi}_i^\top \boldsymbol{V}(T)^\top \boldsymbol{B}_T[i, j].$$

*Proof.* In this proof, we will make the encoding of the indices for the Jacobian $d\boldsymbol{W}(T)/d\boldsymbol{W}_0$ more explicit, as mentioned in [Appendix B](). Recall from [Proposition 2]() that this Jacobian matrix can be decomposed as

$$\frac{d\boldsymbol{W}(T)}{d\boldsymbol{W}_0} = \boldsymbol{I} - \sum_{i,j} \boldsymbol{B}_T[i, j] \otimes \boldsymbol{\phi}_i \boldsymbol{\phi}_j^\top.$$

Introducing the following notations

$$\boldsymbol{F} \triangleq \sum_{i,j} \boldsymbol{B}_T[i, j] \otimes \boldsymbol{\phi}_i \boldsymbol{\phi}_j^\top \in \mathbb{R}^{Nd \times Nd} \qquad \boldsymbol{G} \triangleq \text{Vec}\big(\boldsymbol{V}(T)\big)^\top \boldsymbol{F} \in \mathbb{R}^{Nd},$$

for all $l \in \{0, \dots, N-1\}$ and $y \in \{0, \dots, d-1\}$:

$$
\begin{aligned}
\boldsymbol{G}[dl + y] &= \sum_{k=0}^{N-1} \sum_{x=0}^{d-1} \boldsymbol{V}_T[k, x] \boldsymbol{F}[dk + x, dl + y] \\
&= \sum_{i,j} \sum_{k=0}^{N-1} \Bigg[ \underbrace{\sum_{x=0}^{d-1} \boldsymbol{V}_T[k, x] \boldsymbol{\phi}_i[x]}_{= [\boldsymbol{V}(T)\boldsymbol{\phi}_i]_k} \Bigg] \boldsymbol{B}_T[i, j, k, l] \boldsymbol{\phi}_j[y] \\
&= \sum_{i,j} \Bigg[ \underbrace{\sum_{k=0}^{N-1} [\boldsymbol{V}(T)\boldsymbol{\phi}_i]_k \boldsymbol{B}_T[i, j, k, l]}_{= [\boldsymbol{\phi}_i^\top \boldsymbol{V}(T)^\top \boldsymbol{B}_T[i,j]]_l} \Bigg] \boldsymbol{\phi}_j[y]
\end{aligned}
$$

$$= \sum_{j=1}^{M} \left[ \underbrace{\sum_{i=1}^{M} \left[ \boldsymbol{\phi}_i^{\top} \boldsymbol{V}(T)^{\top} \boldsymbol{B}_T[i,j] \right]_l}_{= \boldsymbol{C}_{j,l}} \right] \boldsymbol{\phi}_j[y] = \left[ \boldsymbol{C}^{\top} \boldsymbol{\phi} \right]_{l,y}$$

This shows that $\boldsymbol{G}$ is equal to $\boldsymbol{C}^{\top} \boldsymbol{\phi}$ up to reshaping. Using the full form of the Jacobian, including $\boldsymbol{I}$, concludes the proof. $\square$

An interesting observation is that if the term $\boldsymbol{C}^{\top} \boldsymbol{\phi}$ is ignored in the computation of the meta-gradient, we recover an equivalent of the first-order approximation introduced in Finn et al. (2017). Similarly, we can show that we can perform the projection onto the Jacobian matrix $d\boldsymbol{W}(T)/d\boldsymbol{\phi}_m$ without having to form the explicit $Nd \times d$ matrix.

**Proposition 6.** *Let $f_{\Phi}(\mathcal{D}_{\tau}^{\mathrm{train}}) = \{(\boldsymbol{\phi}_m, \boldsymbol{y}_m)\}_{m=1}^{M}$ and $f_{\Phi}(\mathcal{D}_{\tau}^{\mathrm{test}})$ be the embedded training set and test set through the embedding network $f_{\Phi}$ respectively. Let $\boldsymbol{s}_m(T)$ be the solution at time $T$ of the differential equation defined in Proposition 1, $\boldsymbol{B}_T[i,j]$ the solution of the one defined in Proposition 2, and $\boldsymbol{z}_T[i,j,m]$ the solution of the one defined in Proposition 3. Let $\boldsymbol{V}(T) \in \mathbb{R}^{N \times d}$ be the partial derivative of the outer-loss wrt. the adapted parameters $\boldsymbol{W}(T)$:*

$$\boldsymbol{V}(T) = \frac{\partial \mathcal{L}\big(\boldsymbol{W}(T); f_{\Phi}(\mathcal{D}_{\tau}^{\mathrm{test}})\big)}{\partial \boldsymbol{W}(T)}.$$

*Let $\boldsymbol{\phi} = [\boldsymbol{\phi}_1, \ldots, \boldsymbol{\phi}_M]^{\top} \in \mathbb{R}^{M \times d}$ be the design matrix. The projection of $\boldsymbol{V}(T)$ onto the Jacobian matrix $d\boldsymbol{W}(T)/d\boldsymbol{\phi}_m$ can be computed as*

$$\mathrm{Vec}\big(\boldsymbol{V}(T)\big)^{\top} \frac{d\boldsymbol{W}(T)}{d\boldsymbol{\phi}_m} = -\big[\boldsymbol{s}_m(T)^{\top} \boldsymbol{V}(T) + \boldsymbol{C}_m \boldsymbol{W}_0 + \boldsymbol{D}_m \boldsymbol{\phi}\big],$$

*where $\boldsymbol{C}_m \in \mathbb{R}^N$ and $\boldsymbol{D}_m \in \mathbb{R}^M$ are defined by*

$$\boldsymbol{C}_m = \sum_{i=1}^{M} \boldsymbol{\phi}_i^{\top} \boldsymbol{V}(T)^{\top} \boldsymbol{B}_T[i,m] \qquad\qquad \boldsymbol{D}_{m,j} = \sum_{i=1}^{M} \boldsymbol{z}_T[i,j,m]^{\top} \boldsymbol{V}(T) \boldsymbol{\phi}_i$$

Note that the definition of $\boldsymbol{C}$ in Proposition 6 matches exactly the definition of $\boldsymbol{C}$ in Proposition 5, and therefore we only need to compute this matrix once to perform the projections for both meta-gradients.

*Proof.* Recall from Proposition 3 that the Jacobian $d\boldsymbol{W}(T)/d\boldsymbol{\phi}_m$ can be decomposed as

$$\frac{d\boldsymbol{W}(T)}{d\boldsymbol{\phi}_m} = -\left[\boldsymbol{s}_m(T) \otimes \boldsymbol{I}_d + \sum_{i=1}^{M} \boldsymbol{B}_T[i,m] \boldsymbol{W}_0 \otimes \boldsymbol{\phi}_i + \sum_{i,j} \boldsymbol{z}_T[i,j,m] \boldsymbol{\phi}_j^{\top} \otimes \boldsymbol{\phi}_i\right].$$

We will consider each of these 3 terms separately.

- For the first term, let

$$\boldsymbol{F}_1 \triangleq \boldsymbol{s}_m(T) \otimes \boldsymbol{I}_d \in \mathbb{R}^{Nd \times d} \qquad\qquad \boldsymbol{G}_1 \triangleq \mathrm{Vec}\big(\boldsymbol{V}(T)\big)^{\top} \boldsymbol{F}_1 \in \mathbb{R}^d,$$

and for $y \in \{0, \ldots, d-1\}$:

$$\boldsymbol{G}_1[y] = \sum_{k=0}^{N-1} \sum_{x=0}^{d-1} \boldsymbol{V}_T[k,x] \boldsymbol{F}_1[dk+x, y] = \sum_{k=0}^{N-1} \sum_{x=0}^{d-1} \boldsymbol{V}_T[k,x] \mathbb{1}(x=y) \big[\boldsymbol{s}_m(T)\big]_k$$

$$= \sum_{k=0}^{N-1} \big[\boldsymbol{s}_m(T)\big]_k \boldsymbol{V}_T[k,y] = \big[\boldsymbol{s}_m(T)^{\top} \boldsymbol{V}(T)\big]_y$$

Therefore, $\boldsymbol{G}_1$ the projection of $\boldsymbol{V}(T)$ onto the first term of the Jacobian matrix is equal to $\boldsymbol{s}_m(T)^{\top} \boldsymbol{V}(T)$.

- For the second term, let

$$\boldsymbol{F}_2 \triangleq \sum_{i=1}^{M} \boldsymbol{B}_T[i,m]\boldsymbol{W}_0 \otimes \boldsymbol{\phi}_i \in \mathbb{R}^{Nd\times d} \qquad \boldsymbol{G}_2 \triangleq \mathrm{Vec}\big(\boldsymbol{V}(T)\big)^{\top}\boldsymbol{F}_2 \in \mathbb{R}^d,$$

and for $y \in \{0, \ldots, d-1\}$:

$$\begin{aligned}
\boldsymbol{G}_2[y] &= \sum_{k=0}^{N-1}\sum_{x=0}^{d-1} \boldsymbol{V}_T[k,x]\boldsymbol{F}_2[dk+x,y] \\
&= \sum_{i=1}^{M}\sum_{l=0}^{N-1}\sum_{k=0}^{N-1}\bigg[\underbrace{\sum_{x=0}^{d-1}\boldsymbol{V}_T[k,x]\boldsymbol{\phi}_i[x]}_{=[\boldsymbol{V}(T)\boldsymbol{\phi}_i]_k}\bigg]\boldsymbol{B}_T[i,m,k,l]\boldsymbol{W}_0[l,y] \\
&= \sum_{i=1}^{M}\sum_{l=0}^{N-1}\bigg[\underbrace{\sum_{k=0}^{N-1}[\boldsymbol{V}(T)\boldsymbol{\phi}_i]_k\boldsymbol{B}_T[i,m,k,l]}_{=\big[\boldsymbol{\phi}_i^{\top}\boldsymbol{V}(T)^{\top}\boldsymbol{B}_T[i,m]\big]_l}\bigg]\boldsymbol{W}_0[l,y] \\
&= \sum_{l=0}^{N-1}\bigg[\underbrace{\sum_{i=1}^{M}\big[\boldsymbol{\phi}_i^{\top}\boldsymbol{V}(T)^{\top}\boldsymbol{B}_T[i,m]\big]_l}_{=\boldsymbol{C}_{m,l}}\bigg]\boldsymbol{W}_0[l,y] = \big[\boldsymbol{C}_m\boldsymbol{W}_0\big]_y
\end{aligned}$$

$\boldsymbol{G}_2$ the projection of $\boldsymbol{V}(T)$ onto the second term of the Jacobian matrix is equal to $\boldsymbol{C}_m\boldsymbol{W}_0$.

- Finally for the third term, let

$$\boldsymbol{F}_3 \triangleq \sum_{i,j} \boldsymbol{z}_T[i,j,m]\boldsymbol{\phi}_j^{\top} \otimes \boldsymbol{\phi}_i \in \mathbb{R}^{Nd\times d} \qquad \boldsymbol{G}_3 \triangleq \mathrm{Vec}\big(\boldsymbol{V}(T)\big)^{\top}\boldsymbol{F}_3 \in \mathbb{R}^d$$

and for $y \in \{0, \ldots, d-1\}$:

$$\begin{aligned}
\boldsymbol{G}_3[y] &= \sum_{k=0}^{N-1}\sum_{x=0}^{d-1} \boldsymbol{V}_T[k,x]\boldsymbol{F}_3[dk+x,y] \\
&= \sum_{i,j}\sum_{k=0}^{N-1}\bigg[\underbrace{\sum_{x=0}^{d-1}\boldsymbol{V}_T[k,x]\boldsymbol{\phi}_i[x]}_{=[\boldsymbol{V}(T)\boldsymbol{\phi}_i]_k}\bigg]\boldsymbol{z}_T[i,j,m,k]\boldsymbol{\phi}_j[y] \\
&= \sum_{i,j}\bigg[\underbrace{\sum_{k=0}^{N-1}[\boldsymbol{V}(T)\boldsymbol{\phi}_i]_k\boldsymbol{z}_T[i,j,m,k]}_{=\boldsymbol{z}_T[i,j,m]^{\top}\boldsymbol{V}(T)\boldsymbol{\phi}_i\in\mathbb{R}}\bigg]\boldsymbol{\phi}_j[y] \\
&= \sum_{j=1}^{M}\bigg[\underbrace{\sum_{i=1}^{M}\boldsymbol{z}_T[i,j,m]^{\top}\boldsymbol{V}(T)\boldsymbol{\phi}_i}_{=\boldsymbol{D}_{m,j}}\bigg]\boldsymbol{\phi}_j[y] = \big[\boldsymbol{D}_m\boldsymbol{\phi}\big]_y
\end{aligned}$$

$\boldsymbol{G}_3$ the projection of $\boldsymbol{V}(T)$ onto the third term of the Jacobian is equal to $\boldsymbol{D}_m\boldsymbol{\phi}$.

$\square$

Note that the projection of $\boldsymbol{V}(T)$ as defined in Proposition 6 is not equal to the meta-gradient itself, as it is missing the term coming from the partial derivative of the outer-loss wrt. the embedding vector $\partial \mathcal{L}\big(\boldsymbol{W}(T); f_\Phi(\mathcal{D}_\tau^{\text{test}})\big)/\partial\boldsymbol{\phi}_m$, which is non-zero unlike the counterpart for $\boldsymbol{W}_0$. However, this projection is the only expensive operation required to compute the total gradient.

## D    PROOF OF STABILITY

In this section, we would like to show that unlike the adjoint method (see Sections 2.2 and 3.2), our method to compute the meta-gradients of COMLN is guaranteed to be *stable*. In other words, even under some small perturbations due to the ODE solver, the solution found by numerical integration is going to stay close to the true solution of the dynamical system. To do so, we use the concept of Lyapunov stability of the solution of an ODE, which we recall here for completeness:

**Definition 1** (Lyapunov stability; Wiggins, 2003). *The solution $\bar{x}(t)$ of a dynamical system is said to be* Lyapunov stable *if, given $\varepsilon > 0$, there exists $\delta$ such that for any other solution $x(t)$ such that $\|\bar{x}(0) - x(0)\| < \delta$, we have $\|\bar{x}(t) - x(t)\| < \varepsilon$ for all $t > 0$.*

We would like to understand the conditions needed for a function $f : \mathbb{R}^n \to \mathbb{R}^n$, such that a trajectory $\bar{x}(t)$ satisfying the following autonomous differential equation is (Lyapunov) stable:

$$\frac{dx}{dt} = f\big(x(t)\big)$$

If we assume for a moment that $f(x) = -\nabla \mathcal{L}(x)$ where $\mathcal{L}$ is a convex function, then for any two trajectories $x_1(t)$ & $x_2(t)$ of the dynamical system above, we can see that the function $F$ defined by

$$F(t) = \frac{1}{2}\|x_1(t) - x_2(t)\|^2$$

satisfies $\dot{F}(t) \leq 0$, where we use the notation $\dot{F}(t) \triangleq dF/dt$. Indeed, since $\nabla^2 \mathcal{L}$ is positive semi-definite and by defining $h(t) = x_2(t) - x_1(t)$, we get

$$\begin{aligned}
\dot{F}(t) = \frac{dF}{dt} &= \left[\frac{dF}{dh}\right]^\top \frac{dh}{dt} \\
&= \big(x_2(t) - x_1(t)\big)^\top \big(\dot{x}_2(t) - \dot{x}_1(t)\big) \\
&= \big(x_2(t) - x_1(t)\big)^\top \big(f(x_2(t)) - f(x_1(t))\big) \\
&= h(t)^\top \left[\int_0^1 Df\big(x_1(t) + sh(t)\big)ds\right] h(t) \\
&= -\int_0^1 \underbrace{h(t)^\top \nabla^2 \mathcal{L}\big(x_1(t) + sh(t)\big)h(t)}_{\geq 0}\, ds
\end{aligned} \qquad (15)$$

where (15) follows from the *fundamental theorem of calculus*. Now since $\dot{F}(t) \leq 0$, the distance between two trajectories decreases as time $t$ increases, hence we have Lyapunov stability for all trajectories or solutions of the above autonomous differential equation.

Now note that we didn't actually need $f$ to be of the specific form $f(x) = -\nabla \mathcal{L}(x)$, but having the Jacobian matrix $Df$ negative semi-definite everywhere would have been sufficient. Hence we get the following statement

**Proposition 7.** *If a continuously differentiable function $f : \mathbb{R}^n \to \mathbb{R}^n$ is such that the Jacobian matrix $Df$ is negative semi-definite everywhere, then any solution $\bar{x}(t)$ of the autonomous differential equation*

$$\frac{dx}{dt} = f\big(x(t)\big)$$

*with initial condition $\bar{x}(0) = x_0$ is (Lyapunov) stable.*

We can extend the above result to the non-autonomous case. For this purpose, let us define

$$\frac{dx}{dt} = f\big(t, x(t)\big) \qquad (16)$$

which induces the parametrized function $f_t : x \mapsto f(t, x)$. With this notation we can state the following result.

**Proposition 8.** *If the Jacobian matrix $Df_t$ is negative semi-definite everywhere for all $t \geq 0$, then any trajectory $x(t)$, solution of (16) with initial condition $x(0) = x_0$, is (Lyapunov) stable.*

*Proof.* Let us start with two trajectories $\boldsymbol{x}_1(t)$ and $\boldsymbol{x}_2(t)$, which are solutions to the above (16). By defining $\boldsymbol{h}(t) = \boldsymbol{x}_2(t) - \boldsymbol{x}_1(t)$ and by applying the *fundamental theorem of calculus*, we get

$$f\big(t, \boldsymbol{x}_1(t)\big) - f\big(t, \boldsymbol{x}_2(t)\big) = f_t\big(\boldsymbol{x}_1(t)\big) - f_t\big(\boldsymbol{x}_2(t)\big)$$

$$= \left[\int_0^1 Df_t\big(\boldsymbol{x}_1(t) + s\boldsymbol{h}(t)\big)ds\right]\boldsymbol{h}(t)$$

Following the same idea as already previously outlined, let us define $F(t) = \frac{1}{2}\|\boldsymbol{x}_1(t) - \boldsymbol{x}_2(t)\|^2$ and show that $\dot{F}(t) \le 0$:

$$\dot{F}(t) = (\boldsymbol{x}_2(t) - \boldsymbol{x}_1(t))^\top \big(f(t, \boldsymbol{x}_2(t)) - f(t, \boldsymbol{x}_1(t))\big)$$

$$= \boldsymbol{h}(t)^\top \left[\int_0^1 Df_t\big(\boldsymbol{x}_1(t) + s\boldsymbol{h}(t)\big)ds\right]\boldsymbol{h}(t)$$

$$= \int_0^1 \underbrace{\boldsymbol{h}(t)^T Df_t\big(\boldsymbol{x}_1(t) + s\boldsymbol{h}(t)\big)\boldsymbol{h}(t)}_{\le 0}\, ds \le 0$$

Hence the distance between two trajectories decreases as time $t$ increases, and we have Lyapunov stability for all solutions of the non-autonomous differential equation above. $\square$

### D.1 STABILITY OF THE FORWARD SENSITIVITY EQUATIONS

In this subsection let us start by considering the general case of solving the initial value problem to the following autonomous system of differential equations:

$$\begin{cases} \dfrac{d\boldsymbol{W}}{dt} = g(\boldsymbol{W}(t), \boldsymbol{\theta}) & \boldsymbol{W}(0) = \boldsymbol{W}_0 \\[2ex] \dfrac{d\mathcal{S}}{dt} = \dfrac{\partial g\big(\boldsymbol{W}(t), \boldsymbol{\theta}\big)}{\partial \boldsymbol{W}(t)} \cdot \mathcal{S}(t) + \dfrac{\partial g\big(\boldsymbol{W}(t), \boldsymbol{\theta}\big)}{\partial \boldsymbol{\theta}} & \mathcal{S}(0) = \dfrac{\partial \boldsymbol{W}(0)}{\partial \boldsymbol{\theta}}. \end{cases} \quad (17)$$

**Proposition 9.** *There exists a solution of the autonomous system of differential equations in (17), which is also unique.*

*Proof.* By Theorem 7.1.1 in Wiggins (2003), we know that there exists a solution of the system of differential equations in Eq. 17. Then, by Theorem 7.4.1 in Wiggins (2003), we can conclude that this solution is unique upon the choice of initial conditions $\boldsymbol{W}(0) = \boldsymbol{W}_0$ and $\mathcal{S}(0) = \partial \boldsymbol{W}(0)/\partial \boldsymbol{\theta}$.

Alternatively, one could also apply Theorems 7.1.1 & 7.4.1 in Wiggins (2003) to conclude that the initial value problem $d\boldsymbol{W}/dt = g(\boldsymbol{W}(t), \boldsymbol{\theta})$ with initial condition $\boldsymbol{W}(0) = \boldsymbol{W}_0$ has a unique solution. This existence and uniqueness then gives rise to existence and uniqueness of a solution of the entire system of differential equations via $\mathcal{S}(t) = d\boldsymbol{W}/dt$ by applying Clairnaut's theorem. $\square$

In our case, we have $g(\boldsymbol{W}(t), \theta) = -\nabla\mathcal{L}\big(\boldsymbol{W}(T), \boldsymbol{\theta}\big)$, where $\mathcal{L}$ is a convex function in $\boldsymbol{W}$. Hence the autonomous system of differential equations can be rewritten as

$$\begin{cases} \dfrac{d\boldsymbol{W}}{dt} = -\nabla\mathcal{L}\big(\boldsymbol{W}(T), \boldsymbol{\theta}\big) & \boldsymbol{W}(0) = \boldsymbol{W}_0 \\[2ex] \dfrac{d\mathcal{S}}{dt} = -\nabla^2\mathcal{L}\big(\boldsymbol{W}(T), \boldsymbol{\theta}\big)\mathcal{S}(t) + \dfrac{\partial g\big(\boldsymbol{W}(t), \boldsymbol{\theta}\big)}{\partial \boldsymbol{\theta}} & \mathcal{S}(0) = \dfrac{\partial \boldsymbol{W}(0)}{\partial \boldsymbol{\theta}}. \end{cases} \quad (18)$$

**Proposition 10.** *Let $\boldsymbol{W}(t)$ be the solution of*

$$\frac{d\boldsymbol{W}}{dt} = -\nabla\mathcal{L}\big(\boldsymbol{W}(T), \boldsymbol{\theta}\big) \qquad\qquad \boldsymbol{W}(0) = \boldsymbol{W}_0$$

*Then the trajectory $\boldsymbol{W}(t)$ is (Lyapunov) stable.*

*Proof.* Note that $Dg\big(\boldsymbol{W}(t)\big) = -\nabla^2\mathcal{L}$ here. Since $\mathcal{L}$ is convex, $\nabla^2\mathcal{L}$ is positive semi-definite, and hence $Dg\big(\boldsymbol{W}(t)\big)$ is negative semi-definite. Hence the result directly follows from Proposition 7. $\square$

Table 4: The effect of the numerical solver on the performance of COMLN. The average accuracy (%) on $1,000$ held-out meta-test tasks is reported with $95\%$ confidence interval. Note that for a given setting, the same $1,000$ tasks are used for evaluation, making both methods directly comparable. RK: 4th-order Runge-Kutta with Dormand Prince adaptive step size. Euler: explicit Euler scheme.

| Method | *mini*ImageNet 5-way | |
| --- | --- | --- |
| | 1-shot | 5-shot |
| COMLN (RK) | $53.01 \pm 0.62$ | $70.54 \pm 0.54$ |
| COMLN (Euler) | $53.00 \pm 0.83$ | $70.50 \pm 0.72$ |

In addition we can now also conclude that the solution of (18) is also (Lyapunov) stable.

**Corollary 1.** *The solution $\big[\boldsymbol{W}(t), \mathcal{S}(t)\big]$ of (18) is (Lyapunov) stable.*

This result is general to the application of the forward sensitivity equations on a gradient vector field derived from a convex loss function. We can also show that when the gradient $d\boldsymbol{W}^\star/d\boldsymbol{\theta}$ exists and is finite (recall that $\boldsymbol{W}^\star$ is the minimizer of the loss function, and the equilibrium of the gradient vector field), then the solution of the system is also bounded, which guarantees us that our solution will not diverge (unlike the adjoint method applied to a gradient vector field).

**Proposition 11.** *Assuming that $\big\|d\boldsymbol{W}^\star/d\boldsymbol{\theta}\big\|$ is finite, the solution $\big[\boldsymbol{W}(t), \mathcal{S}(t)\big]$ of Eq. 18 is bounded.*

*Proof.* Let us define

$$V(t) = \frac{1}{2}\|\boldsymbol{W}(t)\|^2 + \frac{1}{2}\|\mathcal{S}(t)\|^2$$

Since $\boldsymbol{W}(t)$ and $\mathcal{S}(t)$ are continuous functions in $t$, verifying

$$\boldsymbol{W}(t) \underset{t\to\infty}{\longrightarrow} \boldsymbol{W}^\star \qquad\qquad \text{and} \qquad\qquad \mathcal{S}(t) \underset{t\to\infty}{\longrightarrow} \frac{d\boldsymbol{W}^\star}{d\boldsymbol{\theta}}$$

Hence

$$V(t) \underset{t\to\infty}{\longrightarrow} \frac{1}{2}\|\boldsymbol{W}^\star\|^2 + \frac{1}{2}\left\|\frac{d\boldsymbol{W}^\star}{d\boldsymbol{\theta}}\right\|^2 < \infty,$$

where the last point follows from our assumption that $\big\|d\boldsymbol{W}(t)/d\boldsymbol{\theta}\big\| < \infty$. We also have that

$$V(0) = \frac{1}{2}\|\boldsymbol{W}_0\|^2 + \frac{1}{2}\left\|\frac{d\boldsymbol{W}(0)}{d\boldsymbol{\theta}}\right\|^2 < \infty.$$

$V(t)$ being continuous as a composition of continuous functions, we conclude that $V$ is bounded, and hence $\boldsymbol{W}(t)$ and $\mathcal{S}(t)$ are bounded as well. $\qquad\square$

# E EXPERIMENTAL DETAILS

## E.1 CHOICE OF THE NUMERICAL SOLVER

In Section 5.2, we showed that the numerical solver had a significant impact on the time to compute the meta-gradients (but not the memory requirements). In Table 4, we show that using explicit Euler instead of an adaptive scheme like Runge-Kutta leads to very similar results. In all our experiments we therefore chose Runge-Kutta for its faster execution, and by convenience—it is the default numerical solver available in JAX (Bradbury et al., 2018).

## E.2 DETAILS FOR MEASURING THE EFFICIENCY

In Figures 3 & 4, we show the efficiency, both in terms of memory and runtime, of COMLN compared to other meta-learning algorithms. The computational efficiency (runtime) was measured as the average time taken to compute the meta-gradients on a single 5-shot 5-way task from the *mini*ImageNet dataset over 100 runs.

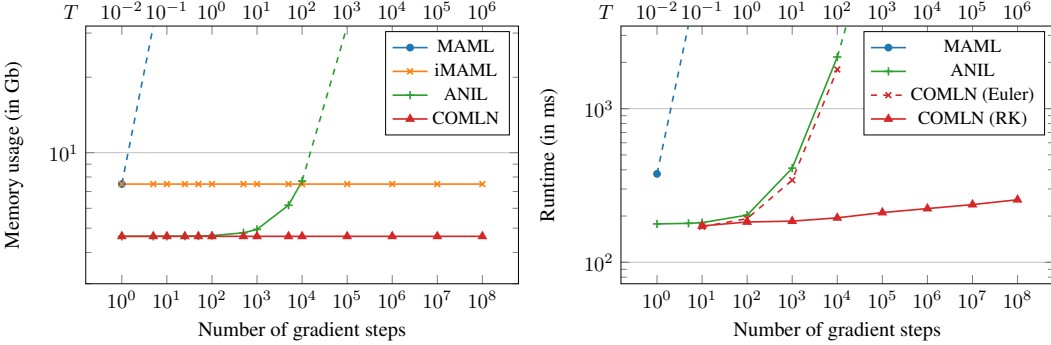

Figure 4: Empirical efficiency of COMLN on a single 5-shot 5-way task, with a ResNet-12 backbone; this figure is similar to Figure 3. (Left) Memory usage for computing the meta-gradients as a function of the number of inner-gradient steps. The extrapolated dashed lines correspond to the method reaching the memory capacity of a Tesla V100 GPU with 32Gb of memory. (Right) Average time taken (in ms) to compute the exact meta-gradients. The extrapolated dashed lines correspond to the method taking over 3 seconds.

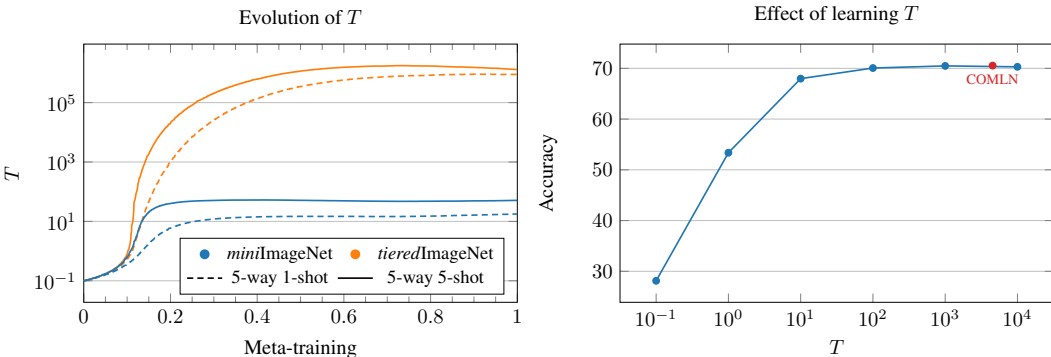

Figure 5: (Left) Evolution of the meta-parameter $T$ controlling the amount of adaptation necessary for all tasks during meta-training. Here, the backbone is a ResNet-12. We normalized the duration of meta-training in $[0, 1]$ to account for early stopping; typically the model for *mini*ImageNet requires an order of magnitude fewer iterations. (Right) Comparison of COMLN (where $T$ is learned, in red) to meta-learning with a fixed length of adaptation $T$ (in blue), on a 5-shot 5-way classification problem on the *mini*ImageNet dataset (with a Conv-4 backbone).

In order to ensure fair comparison between methods that rely on a discrete number of gradient steps (MAML, ANIL, and iMAML), and COMLN which relies on a continuous integration time $T$, we added a conversion between the number of gradient steps and $T$. This corresponds to taking a learning rate of $\alpha = 0.01$ in (2) (which is standard practice for MAML and ANIL on miniImageNet). This means that the number of gradient steps is $100\times$ larger than $T$. This can be formally justified by considering an explicit Euler scheme for COMLN (see Section 3.3), with a constant step size $\alpha = 0.01$. The memory requirements is independent of the choice of the numerical solver. For the computation time, this correspondence between $T$ and the number of gradient steps is no longer exact for *COMLN (RK)*—the comparison with *COMLN (Euler)* is still valid though.

### E.3 ANALYSIS OF THE LEARNED HORIZON

Since one of the advantage of COMLN compared to other gradient-based methods is its capacity to learn the amount of adaptation through the time horizon $T$, Figure 5 (left) shows the evolution of this meta-parameter during meta-training. We can observe that for the more complex dataset *tiered*ImageNet, COMLN appropriately learns to use a longer sequence of adaptation. Similarly

Table 5: *mini*ImageNet results using LEO embeddings and a single linear classifier layer. The average accuracy (%) on 1,000 held-out meta-test tasks is reported with 95% confidence interval. * Results reported in (Rusu et al., 2018). ** Note that LEO uses more than a single linear classifier layer, but we add the numbers for completeness.

| Model | *mini*ImageNet 5-way | |
|---|---|---|
| | 1-shot | 5-shot |
| MAML (Finn et al., 2017) | $50.35 \pm 0.63$ | $65.28 \pm 0.54$ |
| Meta-SGD$^*$ (Li et al., 2017) | $54.24 \pm 0.03$ | $70.86 \pm 0.04$ |
| Meta-SGD (Li et al., 2017) | $50.57 \pm 0.64$ | $69.09 \pm 0.53$ |
| iMAML (Rajeswaran et al., 2019) | $50.26 \pm 0.61$ | $69.52 \pm 0.51$ |
| R2D2 (Bertinetto et al., 2018) | $50.33 \pm 0.62$ | $70.38 \pm 0.52$ |
| LRD2 (Bertinetto et al., 2018) | $50.41 \pm 0.62$ | $70.29 \pm 0.52$ |
| LEAP (Flennerhag et al., 2018) | $50.95 \pm 0.62$ | $66.72 \pm 0.55$ |
| MetaOptNet (Lee et al., 2019) | $40.60 \pm 0.60$ | $50.94 \pm 0.62$ |
| LEO** (Rusu et al., 2018) | $61.76 \pm 0.08$ | $77.59 \pm 0.12$ |
| **COMLN (Ours)** | $50.39 \pm 0.63$ | $70.06 \pm 0.52$ |

within each dataset, it also learns to use shorter sequences of adaptation of 1-shot problems, possibly to allow for better generalization and to reduce overfitting.

Besides adapting the amount of adaptation to the problem at hand, learning $T$ also has the advantage of saving computation while reaching high levels of performance. If we were to fix $T$ ahead of meta-training (as is typically the case in gradient-based meta-learning, where the number of gradient steps for adaptation is a hyperparameter) to a large value in order to reach high accuracy, as is shown in Figure 5 (right), then it would induce larger computational costs early on during meta-training compared to COMLN, which achieves equal performance while tuning the value of $T$. In COMLN, the value of $T$ is relatively small at the beginning of meta-training.

### E.4 ADDITIONAL EXPERIMENT: PREPROCESSED *mini*IMAGENET DATASET

In addition to our experiments on the *mini*ImageNet and *tiered*ImageNet datasets in Section 5, we want to evaluate the performance of continuous-time adaptation in isolation from learning the embedding network. We evaluate this using a preprocessed version of *mini*ImageNet introduced in Rusu et al. (2018), where the embeddings were trained using a Wide Residual Network (WRN) via supervised classification on the meta-train set. For our purposes, we consider these embeddings $\phi \in \mathbb{R}^{640}$ as fixed, and only meta-learn the initial conditions $\boldsymbol{W}_0$ as well as $T$.

The classification accuracies for COMLN and other baselines using pretrained embeddings are shown in Table 5. COMLN achieves comparable or better performance with a single linear classifier layer as other meta-learning methods in both the 5-way 1-shot and 5-way 5-shot classification tasks. The single exception is the 5-way 1-shot result of Meta-SGD from Rusu et al. (2018), which exceeded the performance of COMLN. However, our implementation of Meta-SGD achieved comparable performance to COMLN. This gap in performance is likely due to the additional data used in Rusu et al. (2018) (meta-train and meta-validation splits) during meta-training, as opposed to using only the meta-training set for all other baselines. These results show that isolated from representation learning, all these meta-learning algorithms (either gradient-based or not) perform similarly, and COMLN is no exception.

One notable exception though is MetaOptNet (Lee et al., 2019), where the performance is not as high as the other baselines when the backbone network is not learned anymore—despite often being the best performing model in Table 2. Our hypothesis is that this discrepancy is due to the accuracy of the QP solver used in MetaOptNet, since learning individual SVMs on 1,000 held-out meta-test tasks leads to performance matching all other methods (about 50% for 5-way 1-shot and about 70% for 5-way 5-shot).

### E.5 EXPERIMENTAL DETAILS

For all methods and all datasets, we used SGD with momentum $0.9$ and Nesterov acceleration, with a decreasing learning rate starting at $0.1$ and decreasing according to the schedule provided by Lee et al. (2019). For meta-training, we followed the standard procedure in gradient-based meta-learning, and meta-trained with a fixed number of shots: for example in *mini*ImageNet 5-shot 5-way, we only used tasks with $k = 5$ training examples for each of the $N = 5$ classes. This contrasts with Lee et al. (2019), which uses a larger number of shots during meta-training than the one used for evaluation (e.g. meta-training with $k = 15$, and evaluating on $k = 1$). This may explain the gap in performance between COMLN and MetaOptNet, especially on 1-shot settings. We opted to not follow this decision made by Lee et al. (2019) to ensure a fair comparison with other gradient-based methods, which all used the process described above.

**Conv-4 backbone** We used a standard convolutional neural network with 4 convolutional blocks (Finn et al., 2017). Each block consists of a convolutional layer with a $3 \times 3$ kernel and $64$ channels, followed by a batch normalization layer, and a max-pooling layer with window size and stride $2 \times 2$. The activation function is a ReLU.

**ResNet-12 backbone** We largely followed the architecture from Lee et al. (2019), which consists of a 12-layer residual network. The neural network is composed of 4 blocks with residual connections of 3 convolutional layers with a $3 \times 3$ kernel. The convolutional layers in the residual block have $k = 64, 160, 320$ and $640$ channels respectively. The non-linearity functions are LEAKYRELU$(0.1)$, and a max-pooling layer with window size and stride $2 \times 2$ is applied at the end of each block. No global pooling is performed at the end of the embedding network, meaning that the embedding dimension is $d = 16{,}000$. The only notable difference with the architecture used by Lee et al. (2019) is the absence of DropBlock (Ghiasi et al., 2018) regularization.

