# OpenReview forum: "Continuous-Time Meta-Learning with Forward Mode Differentiation"
_ICLR.cc/2022/Conference — ICLR 2022 Spotlight_

### Official Review · Reviewer_Bwjf · 2021-11-02

**Correctness:** 4
**Technical Novelty And Significance:** 3
**Empirical Novelty And Significance:** 2
**Recommendation:** 8
**Confidence:** 4

**Main Review:**

Strengths:
* The forward-mode differentiation is well-justified. Good discussion on the alternatives and why they are not feasible for the gradient-based meta-learning case.
* The derivation for the vector-Jacobian product in forward-mode differentiation without explicitly constructing the Jacobian is novel and can greatly reduce memory cost in the few-shot setting.
* The adaptation w.r.t. T is interesting, and only possible in the continuous-time formulation. However, I have a few questions about this (see “Questions” section below).

Weaknesses:
* The memory cost scales with the number of data points available. This is not desirable as we would usually expect the performance to improve with more data points, instead of having to limit the amount of data due to constraints on computation resources.


Questions:
* Adapting both T and W0 sounds like two conflicting things to do. When T is adapted to be very large, won’t it drive the gradient w.r.t. W0 to zero (i.e. inner loop is allowed to converge, making W_T independent of initialization)? Does that match your observations in the experiments, and what are the justifications for adapting both? I imagine this conflicting adaptation would complicate the learning dynamics, and may be affected by how you choose the meta optimizers (e.g. if you use a different meta learning rate for T and W0, it might lead to different behaviours).
* What makes COMLN achieve higher accuracy than other gradient-based meta-optimizers? Is it because of the adaptive T? I’d like to see some ablation experiments about this.
* In the related work section, you wrote “Zhang et al. (2021) also introduce a formulation where the adaptation of prototypes follows a gradient vector field, but they finally opt for modeling it as a Neural ODE, possibly due to the challenges of applying the adjoint method we identified in Section 3.2”. From my understanding, it’s precisely due to the numerical instability of the adjoint method that you **don’t use** Neural ODE, and opt for forward-mode differentiation instead. Could you clarify that in more detail?

Feedback for improvements:
* Should further discuss why you choose to focus on gradient-based meta optimizers specifically, what are the pros and cons compared to non-gradient-based ones (like MetaOptNet).

A couple places where \citet should be used instead of \citep:
 * Page 7, third line under “experiments” section: Lee et al., 2019.
 * Page 9, third line of the paragraph before Section 7: Chen et al., 2020a.


**Summary Of The Paper:**

The authors propose COMLN -- a gradient-based meta-learning algorithm with continuous-time inner loop adaptation. The authors discusses possible options for computing the meta-gradient (backpropagation through time, solve ODE backwards in time, and forward-mode differentiation), and identifies the forward-mode differentiation as the only method that is both numerically stable and has a memory cost that is constant w.r.t. the inner-loop length. The authors then derive a way to compute the vector-Jacobian products without explicitly constructing the Jacobians in forward-mode differentiation (specifically for the few-shot learning setting), which further reduces the memory cost. Also, by using a continuous formulation for the inner-optimization, the authors are able to compute meta-gradient w.r.t. horizon T and adapt that as well. Finally, the authors conduct experiments on standard few-shot image classification benchmarks, and show that COMLN outperforms other gradient-based meta-optimization methods in accuracy with a lot less memory cost.

**Summary Of The Review:**

I recommend weak accept. I think the novel derivation of memory-efficient forward-mode differentiation is a significant contribution in the few-shot learning context. I would consider raising the score if my questions about adapting T are addressed.

=== Post rebuttal ===

The authors have addressed all my concerns. I increased the score to 8 (accept).

---

> ### Author Response · Authors · 2021-11-17
> **Response to Reviewer Bwjf**
>
> We would like to thank you for your review.
>
> > _"The adaptation w.r.t. T is interesting, and only possible in the continuous-time formulation."_
>
> We would like to point out that adaptation of $T$ _with SGD_ is only possible with a continuous-time formulation. In the related work, we also mention (Chen et al., 2020), where the number of gradient steps $T$ in gradient-based meta-learning (a discrete quantity) is learned with variational methods.
>
> > _"The memory cost scales with the number of data points available. This is not desirable as we would usually expect the performance to improve with more data points, instead of having to limit the amount of data due to constraints on computation resources."_
>
> We focus here on the few-shot classification setting (Sec. 2), where the number of training points is limited by design. We fully leverage this assumption to devise our memory-efficient algorithm. However, we do not limit the amount of data to fit our assumption: our experiments were conducted on standard settings for few-shot classification. If there are many more data points, this would indeed challenge our assumption, but would probably fall out of the scope of few-shot learning.
>
> > _"Adapting both T and W0 sounds like two conflicting things to do. When T is adapted to be very large, won’t it drive the gradient w.r.t. W0 to zero (i.e. inner loop is allowed to converge, making W_T independent of initialization)? Does that match your observations in the experiments, and what are the justifications for adapting both?"_
>
> Although $T$ grows to large values, it eventually saturates (Figure 5 in App. E.3), meaning that the model benefits from some level of early stopping. After all, we are trying to find a learning rule that is capable of learning a model that can generalize on new test data, and going all the way to convergence on the training data (corresponding to $T \rightarrow \infty$) would not necessarily be the optimal strategy, because of the risk of overfitting. As a consequence, we still observe some dependence of $W(T)$ on $W_{0}$. One of the strength of meta-learning $T$ in COMLN is that the amount of adaptation required (for early stopping) is decided by the model itself, depending on the nature of the problem to solve: some easier problems require less adaptation (e.g. miniImageNet 1-shot 5-way) than harder ones (e.g. tieredImageNet 5-shot 5-way); see Figure 5.
>
> > _"I imagine this conflicting adaptation would complicate the learning dynamics, and may be affected by how you choose the meta optimizers (e.g. if you use a different meta learning rate for T and W0, it might lead to different behaviors)."_
>
> Perhaps surprisingly, we found that experimentally the meta-optimization in COMLN was very stable. However you are absolutely correct that optimization of these two meta-parameters could conflict. What we found helpful was to have a slower update of $T$ compared to $W_{0}$, to allow the model to correct its initialization as $T$ evolves. In practice, we are using an initial learning rate of $10^{-2}$ for $T$ and $10^{-1}$ for $W_{0}$ (both with a learning rate schedule), but we found that COMLN was not overly sensitive to these hyperparameters, beyond the behavior we mentioned above.
>
> > _"What makes COMLN achieve higher accuracy than other gradient-based meta-optimizers? Is it because of the adaptive T? I’d like to see some ablation experiments about this."_
>
> Our main hypothesis for the higher accuracy of COMLN compared to other gradient-based methods is the increased length of adaptation (Sec. 5.1, §2, lines 3-5). We have added in Figure 5 (right) an ablation study of COMLN (where $T$ is learned) compared to having a fixed budget of adaptation $T$, for various values of $T$. As expected, we found that increasing $T$ was leading to better performance. The main advantage of learning $T$, as opposed to having $T$ fixed beforehand, is that if this fixed $T$ is too large, we would waste some time early on during meta-training as opposed to COMLN where $T$ increases over the couse of meta-optimization (at the beginning, $T$ is relatively small). Another advantage, highlighted in Figure 5 (left) is that meta-learning $T$ allows the model to figure out how much adaptation is required, depending on the difficulty of the problem to solve.
>
> ----
> (Chen et al., 2020) Xinshi Chen, Hanjun Dai, Yu Li, Xin Gao, Le Song, _Learning to Stop While Learning to Predict_

---

> > ### Author Response · Authors · 2021-11-17
> > **Response to Reviewer Bwjf (2)**
> >
> > > _"In the related work section, you wrote “Zhang et al. (2021) also introduce a formulation where the adaptation of prototypes follows a gradient vector field, but they finally opt for modeling it as a Neural ODE, possibly due to the challenges of applying the adjoint method we identified in Section 3.2”. From my understanding, it’s precisely due to the numerical instability of the adjoint method that you *don’t use* Neural ODE, and opt for forward-mode differentiation instead. Could you clarify that in more detail?"_
> >
> > We apologize for the confusion. In COMLN, adaptation is based on a gradient vector field, and we do not learn the dynamics of this ODE (only the initial value $W_{0}$): once the loss landscape is known, the gradient vector field is a fixed ODE. Instead, (Zhang et al., 2021) learn an ODE (with a Neural ODE) to adapt their prototypes. In this context, learning the ODE allows more flexibility, which explains why the adjoint method can be applied. In contrast, we motivated COMLN from the perspective of gradient-based meta-learning, where the adaptation is a fixed procedure (gradient descent/following the gradient vector field), and we only learn the initialization.
> >
> > > _"A couple places where \citet should be used instead of \citep"_
> >
> > Thank you for the feedback, we updated our submission to fix those.
> >
> > ----
> > (Zhang et al., 2021) Baoquan Zhang, Xutao Li, Yunming Ye, Shanshan Feng, Rui Ye, _MetaNODE: Prototype Optimization as a Neural ODE for Few-Shot Learning_

---

> > > ### Comment · Reviewer_Bwjf · 2021-11-22
> > > **Thank you for your response**
> > >
> > > I would like to thank the authors for their detailed response. All my concerns are satisfyingly addressed, and I've increased my score.
> > >
> > > I suggest adding the discussion on the conflicting objective and the meta LR selection for T and W0 in an updated version of the paper.

---

> > > > ### Author Response · Authors · 2021-11-25
> > > > **Post rebuttal Response to Reviewer Bwjf**
> > > >
> > > > We are glad that all your concerns have been addressed, and we appreciate that you increased your score. We will make sure that this discussion will be included in the camera-ready version of the paper.

---

### Official Review · Reviewer_Us8H · 2021-11-02

**Correctness:** 4
**Technical Novelty And Significance:** 4
**Empirical Novelty And Significance:** 4
**Recommendation:** 8
**Confidence:** 4

**Main Review:**

The use of continuous time formulation for the inner-loop adaptation step, while also being able to learn the length of time this adaptation should be simulated is an approach that promises to be quite powerful. Combining this with an efficient forward-mode algorithm for learning the parameters of this inner-adaptation, and meta-learning representations is overall a very significant and novel combination.

The writing is clear and the overall algorithm is well explained.
One issue I spotted:
- in eqn. 11, A is not defined

The empirical evaluation of the approach is also thorough, as is the evaluation of computation and memory usage (which the proposed forward-mode approach is targeted towards).

The discussion of the related work seems complete to my knowledge.

The one major weakness of this approach (specifically the efficient forward mode differentiation method) seems to be that it cannot be used for non-cross-entropy losses. Are there alternative ways to achieve the same thing there?

**Summary Of The Paper:**

In this paper, the authors propose a continuous-time formulation for the inner-loop adaptation in a MAML-like setup. Moreover, the authors propose an efficient forward-mode algorithm to learn the parameters of this inner-loop adaptation that is independent of the number of adaptation steps used. The authors show that this works well for a variety of meta-learning problems.

**Summary Of The Review:**

Overall a very strong paper with novel and significant contributions without major flaws.

---

> ### Author Response · Authors · 2021-11-17
> **Response to Reviewer Us8H**
>
> We would like to thank you for your review.
>
> > _"in eqn. 11, A is not defined"_
>
> Apologies for this missing definition, we intentionally deferred the definition of $A$ to the Appendix due to space constraints. We added a reference to App. B.2.
>
> > _"The one major weakness of this approach (specifically the efficient forward mode differentiation method) seems to be that it cannot be used for non-cross-entropy losses. Are there alternative ways to achieve the same thing there?"_
>
> We invite you to look at the general comment above, where we show a similar result for the Mean Squared Error (MSE) and the L2-regularized cross-entropy. We focused on the cross-entropy loss in this paper specifically because it was a non-trivial contribution of particular interest for classification problems (as opposed to the MSE, whose derivation was significantly simpler). While our memory-efficient algorithm based on forward-mode differentiation cannot be applied to all possible loss functions, the MSE and the (regularized) cross-entropy loss cover a large majority of use-cases in practice.

---

> > ### Comment · Reviewer_Us8H · 2021-11-26
> > **Thank you for the response**
> >
> > I would like to thank the authors for their response, and the additional comment on MSE.
> > I am glad to see the paper headed towards acceptance.

---

### Official Review · Reviewer_nRbR · 2021-11-03

**Correctness:** 3
**Technical Novelty And Significance:** 3
**Empirical Novelty And Significance:** 3
**Recommendation:** 6
**Confidence:** 4

**Main Review:**

Strengths:
- The paper is clearly written, well organized and easy to follow
- I believe the method proposed is an interesting addition to the large family of meta-learning algorithms that target the (by-now-classical) few-shot learning setting.
- The possibility of seamlessly treating also the adaptation horizon as a meta-parameter opens up interesting avenues (which however are left as future work)

Weaknesses:
1. I believe that the main weakness of the proposed approach is that it is quite specific to the setting treated in the paper, mainly few-shot classification with a small number of classes, where the loss is (unregularized) cross-entropy. In my view, the main bottleneck is runtime rather than memory, since forward-mode differentiation, not required to store the intermediate states, can always be implemented in a way that is efficient in memory. In time, however, the computation generally grows with the number of meta-parameters. The authors devise a clever decomposition to avoid most of the burden (which, however, still scales with the number of examples and classes), but fail to discuss the limitations of this approach (regarding the efficiency of the gradient computation). E.g., what if $\mathcal{L}$ is a generic function of the parameters? what if at the base level one wants to adapt also the rest of the parameters of the feature extractor or the bias of a regularizer [1]? In my view, the method remains interesting notwithstanding these potential limitations, but limitations should be clearly mentioned and discussed. Furthermore, I would like to see included in the paper also a runtime column in Table 1, to give a more complete picture of the complexity of the algorithms.
2. In my view, one of the most interesting novelties of this work is the possibility of treating the adaptation time as a meta-parameter. I believe that the paper in its current stage misses on the opportunity of exploiting this. On the one side, this is because of the lack of novelty in the experimental validation, on the other side, this could be potentially linked to the limitations of the method: longer adaptation would be extremely beneficial in the presence of tasks with a different number of examples, however, since the memory (but also the runtime I believe) scales with the number of example, the complexity of  COMLN could be too high in these settings. Could the authors comment on this?
3. Related work could be better mentioned. For instance, the idea of only learning base classifiers is also present in [2,3,4] and not only in ANIL. Furthermore, I believe that the distinction between so-called gradient-based and non-gradient based methods is misleading since also MetaOptNet uses gradients (the difference being that there they have a closed-form expression for those, but this is not at all dissimilar to iMAML).

Minor comments:
4. Clarity: Missing formal definition of the sensitivity state $\mathcal{S}(t)$. The loss $\mathcal{L}$ is used with different inputs: cf (2) and (8).

[1] Denevi, Giulia, et al. "Learning-to-learn stochastic gradient descent with biased regularization." International Conference on Machine Learning. PMLR, 2019.
[2] Franceschi, Luca, et al. "Bilevel programming for hyperparameter optimization and meta-learning." International Conference on Machine Learning. PMLR, 2018.
[3]  Bertinetto, Luca, et al. "Meta-learning with differentiable closed-form solvers." ICLR, 2019.
[4]  Lee, Kwonjoon, et al. "Meta-learning with differentiable convex optimization." Proceedings of the IEEE/CVF Conference on Computer Vision and Pattern Recognition. 2019.

--------------------------------------------------
**Post rebuttal:** I thank the authors for their reply. I appreciate the derivations for the MSE and the regularized CE and the clarifications regarding the extensibility to the general case. I hope that these comments will be included in the final version. On the other hand, I am still convinced that including runtimes in Table 1 is quite important to convey a clearer picture (you could add 2+ lines for COML depending on which ODE solver is used). Regarding standard forward mode, please note that since we are discussing asymptotic behaviour the asymptotic runtime complexity of the two implementations is the same, while clearly, one implementation is more memory efficient than the other (this does not mean that the actual GPU/CPU runtime will be the same, but again, Table 2 reports $O$'s..)
Furthermore, I still believe this work misses to capitalize on its main novelty, i.e. meta-learning task-dependent $T$'s, leaving somewhat the doubt that this is either impractical, or does not bring noticeable advantages. For these reasons, while I am overall positive about this work, I will leave my score unvaried.

**Summary Of The Paper:**

The paper describes an algorithm called COMLN for few-shot meta-learning.
In COMLN, the base level loop is modelled as a continuous-time autonomous ODE that is the gradient vector field of the inner loss.
The authors restrict their discussion to a meta-learning setting where the only base-level parameters are the parameters of a linear classifier. The meta-parameters are the parameters of a neural network that acts as feature extractor, the initialization point for adaptation of the classifier and the adaptation time $T$.
The authors point out that, crucially, it is possible to treat $T$ as a learnable meta-parameter thanks to the continuous reformulation of the base level.
Meta-training is carried out by gradient descent on the meta-parameter, where the gradients are computed with an efficient implementation (in few-shot classification, with a small number of classes) forward mode differentiation that exploits the loss structure.
The authors present experiments on standard benchmark datasets, showing that COMLN achieves better performances than other comparable gradient-based methods.

**Summary Of The Review:**

The paper is well-crafted and the method presented is an interesting addition to the few-shot meta-learning literature, with possibly some limitations that are not well discussed. Furthermore, the paper misses exploiting what probably is the main advantage of the method (meta-learning task-specific, or conditional, $T$'s). These two issues currently limit my score to a weak accept.

---

> ### Author Response · Authors · 2021-11-17
> **Response to Reviewer nRbR**
>
> We would like to thank you for your review.
>
> > _"I believe that the main weakness of the proposed approach is that it is quite specific to the setting treated in the paper, mainly few-shot classification with a small number of classes, where the loss is (unregularized) cross-entropy."_
>
> To clarify, our approach is indeed specific to few-shot classification, however we did not make any assumption on a small number of classes. We followed standard benchmarks from the few-shot learning literature, and experimented on 5-way classification problems. The one assumption we take advantage of though is that the number of training examples is small, which is indeed the case in few-shot learning.
>
> Regarding the limitation to unregularized cross-entropy, we would like to thank you for the suggestion. We invite you to look at the general comment where we show that our method can be extended to the L2-regularized cross-entropy (in addition to the MSE).
>
> > _"In my view, the main bottleneck is runtime rather than memory, since forward-mode differentiation, not required to store the intermediate states, can always be implemented in a way that is efficient in memory. In time, however, the computation generally grows with the number of meta-parameters."_
>
> While runtime is a concern, memory is the largest bottleneck in gradient-based meta-learning (GBML) methods, as we argue in Sec. 4.2 & Table 1: in GBML, the memory requirements typically scale with the length of adaptation $T$, and since in practice we are bounded by the memory limitations of the hardware (GPU), we cannot apply those methods to large $T$s. We show that this is empirically the case in Figures 3 & 4: GBML methods like MAML and ANIL have memory requirements that increase with $T$, while it remains constant for COMLN (and it returns the exact meta-gradients, unlike iMAML).
>
> We would like to also highlight that applying naively forward-mode differentiation would be impractical, as shown in Table 1 and Sec. 4 (§1), since it would scale quadratically with the number of meta-parameters. Our approach leverages the structure of the cross-entropy loss, and the fact that we are in a few-shot learning setting to circumvent this issue. In that sense, it is incorrect that forward-mode differentiation can always be implemented in a memory-efficient way.
>
> To be completely thorough, an alternative approach that aligns with your comment would be to build the meta-gradients component by component using forward-mode differentiation with Jacobian-vector products. This would require us to perform as many forward passes (i.e. full adaptations on the same task) as there are meta-parameters, every pass giving the value of a single component of the meta-gradient. This approach would be impractical because the runtime would scale up significantly. In contrast, the algorithm in COMLN only requires a single pass to compute all the meta-gradients.
>
> > _"what if $\mathcal{L}$ is a generic function of the parameters? what if at the base level one wants to adapt also the rest of the parameters of the feature extractor or the bias of a regularizer [1]?"_
>
> We believe that it would be impossible to devise a similar memory-efficient algorithm to compute the _exact_ meta-gradients whether we are using a generic loss function, or if we want to adapt the feature extractor. However, an exciting line of research that can come out of COMLN is to devise methods based on forward-mode differentiation, and decompositions similar to the ones derived in this paper, to find an approximation of the meta-gradients under these weaker assumptions (e.g. layer-wise decompositions if we want to adapt the whole network, including the feature extractor).
>
> > _"I would like to see included in the paper also a runtime column in Table 1, to give a more complete picture of the complexity of the algorithms."_
>
> Since the runtime depends on the numerical solver used to find the necessary components to compute the meta-gradients, it would be impossible unfortunately to add a runtime column to Table 1. If we use the explicit Euler scheme, then the runtime for all methods would scale with the length of adaptation $T$ (since we at least need to run adaptation), as verified experimentally in Figures 3 & 4. However if we use a different solver, such as Runge-Kutta with an adaptive step-size, then the runtime can be reduced significantly, since it requires fewer function evaluations.
>
> > _"On the one side, this is because of the lack of novelty in the experimental validation"_
>
> To make it clear, our goal was to study the well-established problem of few-shot classification, and not to introduce a novel problem setting.

---

> > ### Author Response · Authors · 2021-11-17
> > **Response to Reviewer nRbR (2)**
> >
> >
> > > _"longer adaptation would be extremely beneficial in the presence of tasks with a different number of examples, however, since the memory (but also the runtime I believe) scales with the number of example, the complexity of COMLN could be too high in these settings. Could the authors comment on this?"_
> >
> > > _"Furthermore, the paper misses exploiting what probably is the main advantage of the method (meta-learning task-specific, or conditional, $T$'s)."_
> >
> > We absolutely share your enthusiasm for learning task-specific $T$s. This was intentionally left as future work though, because this paper is already making a number of technical contributions (the continuous-time meta-learning formulation, the capacity to treat $T$ as a meta-parameters, and the algorithm based on forward-mode differentiation), and we think that learning task-specific $T$s deserves its own paper rather than relegating it to the Appendix of this paper–because of space limitations.
> >
> > With task-specific $T$s, our hypothesis is that more difficult tasks (e.g. either tasks with a larger number of examples as you correctly said, or because the task is inherently more difficult–like classifying different breeds of dogs as opposed to classifying dogs from cats) may require longer adaptation with larger values of $T$. This would support our observation in Figure 5 & App. E.3 that harder problems (1-shot learning, or tieredImageNet) require larger values of $T$.
> >
> > > _"For instance, the idea of only learning base classifiers is also present in [2,3,4] and not only in ANIL."_
> >
> > We mentioned ANIL as one of the works specifically in _gradient-based meta-learning_ where the feature extractor (or part of it) is shared (Sec. 6, §1, lines 3-6). We do mention these other meta-learning methods where the base classifier is shared across tasks under the umbrella of "metric-based meta-learning" (Sec. 6, §1, lines 8-10).
> >
> > > _"Furthermore, I believe that the distinction between so-called gradient-based and non-gradient based methods is misleading since also MetaOptNet uses gradients (the difference being that there they have a closed-form expression for those, but this is not at all dissimilar to iMAML)."_
> >
> > In this paper, we call gradient-based meta-learning specifically the methods that follow the template of Sec. 2.1, that is methods whose adaptation is based on gradient descent. From this point of view, MetaOptNet would not be considered a gradient-based meta-learning algorithm since adaptation is based on a QP-solver, and not gradient descent.
> >
> > > _"Minor comments: 4. Clarity: Missing formal definition of the sensitivity state $\mathcal{S}(t)$. The loss $\mathcal{L}$ is used with different inputs: cf (2) and (8)."_
> >
> > The sensitivity state $\mathcal{S}(t)$ is defined in the text right above Equation 5. For the loss, we are using the same notation $\Phi$ in Equation 2 to denote the task-agnostic meta-parameters, and in Equation 8 for the feature-extractor (also task-agnostic meta-parameters); this should hopefully be clear from the context, apologies for the confusion.

---

> ### Author Response · Authors · 2021-11-25
> **Post rebuttal Response to Reviewer nRbR**
>
> We appreciate your post-rebuttal response, and we respect your decision to maintain your score. The comments in our rebuttal will indeed be included in the camera-ready version of the paper.
>
> To avoid any ambiguity, we would like to respond to your post-rebuttal comment.
>
> > _"On the other hand, I am still convinced that including runtimes in Table 1 is quite important to convey a clearer picture (you could add 2+ lines for COML depending on which ODE solver is used)."_
>
> To reiterate what we said in our response, including (theoretical) runtimes of COMLN would be impossible, unless for Euler discretization with a constant step size. For Runge-Kutta with an adaptive step size, the number of function evaluations (here, evaluations of $\nabla \mathcal{L}(W)$, controlling the runtime of the algorithm) depends heavily on the dynamics of the ODE, which is driven by $\nabla \mathcal{L}(W; f_{\Phi}(D_{\mathrm{train}}))$, which itself depends on both the data $D_{\mathrm{train}}$ (problem dependent) and the feature extractor $f_{\Phi}$ (evolves during meta-training).
>
> We believe that an empirical evaluation of the runtime on a real problem, such as the ones shown in Figures 3 & 4, should give a clear enough picture of how fast COMLN is compared to other meta-learning methods (including with different choices of numerical solvers for COMLN).
>
> > _"Regarding standard forward mode, please note that since we are discussing asymptotic behaviour the asymptotic runtime complexity of the two implementations is the same, while clearly, one implementation is more memory efficient than the other (this does not mean that the actual GPU/CPU runtime will be the same, but again, Table 2 reports $O$'s..)"_
>
> The runtime complexity depends on the number of function evaluations, but the runtime of each function evaluation itself depends on how big the state of the ODE is. This means that even if memory wasn't an issue, the runtime of naive forward sensitivity (whose state is the full Jacobian) would be higher than COMLN (where the state is much smaller). Provided we could give runtimes (e.g. with Euler discretization), the size of the state would be included in the $O$ notation.
>
> > _"Furthermore, I still believe this work misses to capitalize on its main novelty, i.e. meta-learning task-dependent $T$'s, leaving somewhat the doubt that this is either impractical, or does not bring noticeable advantages."_
>
> To reiterate what we said in our response, meta-learning task-specific $T$s was left intentionally as future work. To make it perfectly clear, we have no result yet that would suggest that this extension is impractical or with no noticeable advantage.
>
> Meta-learning task-specific $T$s also presents new technical challenges (in addition to our contributions in this paper), in particular how to represent the task to feed into the network that would return $T(\tau)$. While we do have some ideas on how to proceed, a description of these design choices, as well as a full empirical analysis of the importance of meta-learning $T$s would deserve its own separate paper.
>
> ----
>
> We would be happy to continue this discussion if you feel any of these points require further clarification.

---

### Official Review · Reviewer_PyBo · 2021-11-05

**Correctness:** 4
**Technical Novelty And Significance:** 3
**Empirical Novelty And Significance:** 3
**Recommendation:** 8
**Confidence:** 3

**Main Review:**

Pros:
  - Interesting use of forward-mode differentiation and structure of the gradients to compute meta-gradients efficiently.
  - Learning T as an early stopping is pretty interesting, and T will be increased if the train and test gradients are aligned, and decreased otherwise. (IMO, the authors can expand on interpreting this alignment a bit more after Eq 13.)

Cons:
  - IMO, the claims/explanations for continuous-time and forward-mode differentiation should be separated. SGD simply corresponds to an Euler discretization, and the same forward-mode tricks can be applied just as readily in the discrete-time setting as the continuous-time setting. Perhaps the only strictly continuous-time-related contribution is being able to learn T.
  - I don't have any major worries, but see my comments below.

Comments:
  - "Gradient-based Hyperparameter Optimization Over Long Horizon" discusses efficient forward-mode differentiation for discrete-time meta-learning. which should be worth discussing.
  - Since forward-mode can also be applied in the discrete-time setting, the main motivation for continuous-time seems to be being able to learn T. However, I'm not entirely convinced about its usefulness:

1. The gradient wrt T is very local.
(i) How much does T change over the course of training? Is the learning useful or do you have to initialize it very well?
(ii) How stable is the optimization? For instance, if you were to initialize with a very large or very small T value, is the gradient sufficient for learning a reasonable value for T?

2. The interpretation of T:
(i) Often meta-learning methods (for instance MAML) motivate the use of a fixed number of iterations to learn good meta-parameters that are useful even with a low compute budget (small number of gradient descent iterations). However, in the continuous-time version, changing T changes the amount of computation. If you have a fixed compute cost, would learning T still be useful? This might be a fairer comparison to discrete-time methods as well.

 - Can you explain why COMLN is more expensive with the Euler integration scheme compared to RK? Are you using different step sizes for these methods?

**Summary Of The Paper:**

This paper proposes using a continuous-time formulation along with forward-mode differentiation to perform meta-learning. The setup follows that of ANIL: the weights of the last layer is part of the inner loop optimization while the initialization of these last-layer weights and the rest of the weights are treated as meta-parameters. The proposed method relies on this setup in order to efficiently compute derivatives, which the authors note have a low-rank structure that can be exploited. Furthermore, the weights do not need to be solved over time, and instead a much smaller vector can be solved instead, which is pretty neat.

**Summary Of The Review:**

The approach is interesting; the main novelty is being able to train T, kind of like a learning early stopping. However, T also increases compute cost, which may not be a completely fair comparison to fixed-cost discrete-time approaches. Another minor concern is the lack of control for compute cost due to the training of T.

---

> ### Author Response · Authors · 2021-11-17
> **Response to Reviewer PyBo**
>
> We would like to thank you for your review.
>
> > _"Learning T as an early stopping is pretty interesting, and T will be increased if the train and test gradients are aligned, and decreased otherwise. (IMO, the authors can expand on interpreting this alignment a bit more after Eq 13.)"_
>
> We also found interesting that the meta-gradient wrt. $T$ involved the alignment between train and test gradients, and that this also appeared in the context of multi-task and meta-learning. Our interpretation of this is that the model requires more adaptation as long as it can make more progress in terms of performance on the test data (the gradients are aligned) for at least some tasks. The model stop requiring more adaptation as soon as there is no more progress to be made on average for all tasks (gradients are orthogonal), which exactly corresponds to early stopping.
>
> > _"IMO, the claims/explanations for continuous-time and forward-mode differentiation should be separated. SGD simply corresponds to an Euler discretization, and the same forward-mode tricks can be applied just as readily in the discrete-time setting as the continuous-time setting. Perhaps the only strictly continuous-time-related contribution is being able to learn T."_
>
> We agree that the same derivations can be applied just as well in discrete-time. We opted for a general treatment in continuous-time of our memory-efficient algorithm based on forward-mode differentiation, since the discrete time version (gradient descent) would be a special case of this with Euler discretization. We also highlighted this connection between gradient descent (and the meta-leaarning algorithm associated to it, ANIL) and COMLN in Sec. 3.3.
>
> In terms of continuous-time-related contributions, they include the ability of learning $T$, but also the formulation itself of COMLN (meta-learning with adaptation using a gradient vector field, Equations 7 & 8), as well as the insights about applying the adjoint method to this problem (which would be the natural choice, Sec. 3.2). We tried to seperate as well as possible the contributions for continuous-time meta-learning (Sec. 3) from the contributions for how to compute the exact meta-gradients efficiently using forward-mode differentiation (Sec. 4).
>
> > _""Gradient-based Hyperparameter Optimization Over Long Horizon" discusses efficient forward-mode differentiation for discrete-time meta-learning. which should be worth discussing."_
>
> This work is absolutely relevant, and we would like to thank you for pointing us to this work that we missed during our review for related work. We added this reference to the related work, but unfortunately we couldn't expand on a discussion due to space limitations. We leave the discussion here for reference.
>
> (Micaelli & Stokey, 2021) use forward-mode differentiation to optimize the hyperparameters of SGD with momentum (e.g. learning rate, momentum). These hyperparameters are low-dimensional in nature, and thus forward-mode differentiation is an effective strategy to compute the hyper-gradients. In contrast, our work meta-learns the initialization of the last layer of a neural network, as well as the feature extractor, which are high-dimensional objects, and therefore naively applying forward-mode differentiation to compute the meta-gradients would be impractical. We decompose the Jacobian matrices into smaller objects that follow very specific dynamics to get an efficient algorithm.
>
> > _"The gradient wrt T is very local. (i) How much does T change over the course of training? Is the learning useful or do you have to initialize it very well?"_
>
> T changes significantly over the course of meta-training. See Figure 5 in App. E.3 for the evolution of T over the course of meta-training. We always initialized $T=0.1$, and $T$ could grow as large as $10^{6}$ at the end of meta-training. An interesting observation is that $T$ eventually saturates and does not grow indefinitely, meaning that some level of early-stopping is favorable. The learning of $T$ is useful for leaving the algorithm the option to choose how much adaptation is required to solve a certain class of problems; see the comparison between settings in Figure 5. In terms of performance, learning $T$ is also reaching high performance compared to fixed values of $T$ (see Figure 5, right), without having to tune the fixed value of $T$.
>
> ----
> (Micaelli & Stokey, 2021) Paul Micaelli, Amos Storkey, _Gradient-based Hyperparameter Optimization Over Long Horizons_

---

> > ### Author Response · Authors · 2021-11-17
> > **Response to Reviewer PyBo (2)**
> >
> >
> > > _" (ii) How stable is the optimization? For instance, if you were to initialize with a very large or very small T value, is the gradient sufficient for learning a reasonable value for T?"_
> >
> > For initialization, we opted for $T=0.1$ to start in a setting close to "multi-task learning", where there would be almost no adaptation at all ($T$ very small), and then leave the algorithm the option to perform more and more adaptation along meta-training. We also experimented with smaller initializations of $T$, and the results were largely similar, the only difference being that progress was slower early on during meta-training, since we are using an exponential parametrization of $T$. Overall in all our experiments, the optimization was stable.
> >
> > > _"(i) Often meta-learning methods (for instance MAML) motivate the use of a fixed number of iterations to learn good meta-parameters that are useful even with a low compute budget (small number of gradient descent iterations). However, in the continuous-time version, changing T changes the amount of computation. If you have a fixed compute cost, would learning T still be useful? This might be a fairer comparison to discrete-time methods as well."_
> >
> > Typically in gradient-based meta-learning, the limitation is not necessarily compute but memory. Methods like MAML usually only use a fixed (and small) number of gradient steps because of a memory budget, and not necessarily a compute budget. This motivated our work for a memory-efficient algorithm that would not scale with the amount of adaptation.
> >
> > However increasing $T$ over the course of meta-training does increase the runtime of adaptation. We have conducted an empirical study of this effect in Figures 3 & 4, and we showed that in terms of runtime, with $T \sim 10^{6}$–the order of magnitude $T$ reaches after meta-optimization on harder problems (Figure 5)–COMLN has an equivalent cost to running MAML with 10 steps of gradient descent (and ANIL with $10^{3}$ steps). This runtime depends on the ODE solver though, and using an Euler discretization has a cost equivalent to ANIL.
> >
> > In terms of performance, one likely explanation for better performance is the increased length of adaptation (Sec. 5.1, §2, lines 3-5). If we were to fix a large number of gradient steps ahead of time though, we would waste more time compared to COMLN since at the beginning of meta-optimization, $T$ is relatively small, and increases over the course of meta-training.
> >
> > > _"Can you explain why COMLN is more expensive with the Euler integration scheme compared to RK? Are you using different step sizes for these methods?"_
> >
> > The explicit Euler scheme is using a fixed step size (in our experiments, $\alpha=0.01$ to match prior work). In contrast, the Runge-Kutta solver uses an adaptive step size, meaning that we can take larger steps during adaptation, leading to fewer function evaluations, hence a reduced runtime. We should note that Runge-Kutta does not require setting a step size ahead of time, and these step sizes are determined based on the numerical tolerance allowed (for tolerance, we largely use the default values provided by the default ODE solver in JAX).

---

### Author Response · Authors · 2021-11-17
**General comment**

We would like to thank all the reviewers for their reviews, and we are pleased to see this enthusiasm for our work.

In this comment, we want to address one point in particular that might be of general interest for all the reviewers.

----

## Cross-entropy loss

Reviewers **nRbR** & **Us8H** highlighted that COMLN was designed specifically for the cross-entropy loss. The reason for this choice is twofold:
 - The cross-entropy is widely used as the loss function of choice for classification problems (Sec. 4, §1), which is the setting we study in this paper (Sec. 2, §1).
 - We leverage the structure of the cross-entropy loss to devise our memory-efficient algorithm based on forward-mode differentiation (Sec. 4.1, §1).

We see this second point as a strength of our work. In fact, short on any assumption on the structure, we believe that it would probably be impossible to design a method for a generic loss function that matches the memory-efficiency of COMLN in general. That being said, it is possible to adapt the results from this paper for the **mean squared error (MSE)** (typically used on regression problems), as well as for the **L2-regularized cross-entropy** loss—following a suggestion from Reviewer **nRbR**. We sketch out these extended results here.

### Mean-squared error (MSE)

Recall that we are in a setting where only the last layer of the neural network is adapted. The MSE can be written as

$$ \mathcal{L}(W) = \frac{1}{2M}\sum_{m=1}^{M}\\|W\phi_{m} - y_{m}\\|_{2}^{2}. $$

The key property of the MSE is that its Hessian matrix is independent of $W$, unlike the Hessian of the cross-entropy loss (see Lemma 1 in Appendix B): $ \nabla^{2}\mathcal{L}(W) = \frac{1}{M}\sum_{m=1}^{M}I_{N} \otimes \phi_{m}\phi_{m}^{\top}. $ In particular, the LHS of the $\otimes$ here is a diagonal matrix, which simplifies things considerably. For example, the Jacobian matrix $dW(t)/dW_{0}$ can be decomposed as

$$ \frac{dW(t)}{dW_{0}} = I - \sum_{i,j} B_{t}[i, j]I_{N} \otimes \phi_{i}\phi_{j}^{\top}, $$

where $B_{t}[i, j] \in \mathbb{R}$ is now a scalar, instead of a full $N \times N$ matrix in the case of the cross-entropy loss; this decomposition is very close to the one in Equation 10. The dynamics of $B(t)$ (as an $M \times M$ matrix) are given by the following linear system of equations:

$$ \frac{dB(t)}{dt} = \frac{1}{M}I - GB(t), $$

where $G$ is the Gram matrix of the embeddings $\\{\phi_{m}\\}_{m=1}^{M}$. Note that since this is a linear ODE with constant coefficients, we could solve it using the matrix exponential (instead of using a numerical solver, which is necessary for the cross-entropy loss).

### L2-regularized cross-entropy loss

Taking inspiration from iMAML (Rajeswaran et al., 2019), we also consider the cross-entropy loss regularized by a proximal term around $W_{0}$. This can be written as

$$ \widetilde{\mathcal{L}}(W) = \mathcal{L}(W) + \frac{\lambda}{2}\\|W - W_{0}\\|_{2}^{2}, $$

where $\mathcal{L}(W)$ is the cross-entropy loss (see Lemma 1). In that case, we can adapt Proposition 2 (in Appendix B.2) & Proposition 3 (in Appendix B.3) and their proofs to incorporate this new regularization term. In particular, the Jacobian matrix $dW(t)/dW_{0}$ can be decomposed exactly like Equation 10, as

$$ \frac{dW(t)}{dW_{0}} = I - \sum_{i,j} B_{t}[i, j] \otimes \phi_{i}\phi_{j}^{\top}, $$

where $B_{t}[i, j]$ follows slightly different dynamics than the ones in Equation 11, with an additional $\lambda B_{t}[i, j]$ term:

$$ \frac{dB_{t}[i, j]}{dt} = 1(i = j)A_{i}(t) - A_{i}(t)\sum_{m=1}^{M}(\phi_{i}^{\top}\phi_{m})B_{t}[m, j]\ {\color{red} -\ \lambda B_{t}[i, j]} \qquad B_{0}[i, j] = 0. $$

A similar additional term also appears in the dynamics of $s_{m}(t)$ and $z_{t}[i, j, m]$ used to define the Jacobian matrices $dW(t)/d\phi_{m}$; the form of the decomposition (Equation 12) also remains unchanged though.

----

(Rajeswaran et al., 2019) Aravind Rajeswaran, Chelsea Finn, Sham Kakade, Sergey Levine, _Meta-Learning with Implicit Gradients_

---

### Decision · Program_Chairs · 2022-01-20

**Decision:**

Accept (Spotlight)

**Comment:**

This paper addresses a continuous-time formulation of gradient-based meta-learning (COMLN) where the adaptation is the solution of a differential equations. In general, outer loop optimization requires backpropagating over trajectories involving gradient updates in the inner loop optimization. It is claimed that one of main advantages of COMLN is able to compute the exact meta-gradients in a memory-efficient way, regardless of the length of adaptation trajectory. To this end, the forward-mode differentiation is used, with exploiting the Jacobian matrix decomposition. All the reviewers agree that the derivation of memory-efficient forward-mode differentiation is a significant contribution in the few-shot learning. The paper is well written and has interesting contributions. Authors did a good job in responding to reviewers’ comments during the discussion period. What is missing in this paper is the discussion of some limitations of the proposed method.  This can be improved in the final version. All reviewers agree to champion this paper. Congratulations on a nice work.